# Stochastic Interpolants via Conditional Dependent Coupling

## Abstract

Existing image generation models face critical challenges regarding the trade-off between computation and fidelity. Specifically, models relying on a pretrained Variational Autoencoder (VAE) suffer from information loss, limited detail, and the inability to support end-to-end training. In contrast, models operating directly in the pixel space incur prohibitive computational cost. Although cascade models can mitigate computational cost, stage-wise separation prevents effective end-to-end optimization, hampers knowledge sharing, and often results in inaccurate distribution learning within each stage. To address these challenges, we introduce a unified multistage generative framework based on our proposed **Conditional Dependent Coupling** strategy. It decomposes the generative process into interpolant trajectories at multiple stages, ensuring accurate distribution learning while enabling end-to-end optimization. Importantly, the entire process is modeled as a single unified Diffusion Transformer, eliminating the need for disjoint modules and also enabling knowledge sharing. Extensive experiments demonstrate that our method achieves both high fidelity and efficiency across multiple resolutions.

## 1 Introduction

Generative models have achieved remarkable progress in recent years, driving advances in diverse domains such as natural language processing OpenAI et al. (2024), computer vision Geng et al. (2025a), and scientific modeling Fotiadis et al. (2024). Unlike discrete data generation (e.g., large language models for text), high-dimensional continuous data generation, such as image synthesis, suffers from high complexity and computational demands Rombach et al. (2022); Peebles & Xie (2023a); Ma et al. (2024). The primary challenge lies in balancing fidelity, efficiency, and scalability when learning to approximate intricate data distributions Sun et al. (2024); Tian et al. (2024b).

There are two main paradigms for generative modeling: pixel space generation and latent space generation. Pixel space generation operates directly in the original data domain, preserving fine-grained details without compression Bao et al. (2023); Hoogeboom et al. (2025). However, this approach suffers from extreme inefficiency due to the high dimensionality of images, resulting in expensive training and inference Rombach et al. (2022); Kingma et al. (2023). In contrast, latent space generation leverages compact representations, typically through Variational Autoencoders (VAEs), to reduce dimensionality and improve efficiency Li et al. (2024); Esser et al. (2021a). While effective, these methods inevitably incur information loss during encoding and decoding, and often cannot be trained in a fully end-to-end manner with the generative backbone Jin et al. (2024); Jiao et al. (2025).

To mitigate these limitations, multi-stage generation methods have been proposed. By decomposing the synthesis process into a sequence of stages, they allow early stages to operate in lower-dimensional spaces and later stages to refine the outputs at higher resolutions Tian et al. (2024b); Chen et al. (2025). This hierarchical design improves efficiency and progressively captures data complexity Li et al. (2025); Ren et al. (2024). Nonetheless, existing multi-stage frameworks rely on disentangled stage-specific models, preventing unified parameterization and end-to-end optimization Ho et al. (2022b); Kim et al. (2024). Furthermore, the decoupled design may lead to inaccurate distribution modeling, with errors compounding across stages Chen et al. (2025); Jin et al. (2024).

In this work, we propose a novel multi-stage generation framework based on *stochastic interpolant* Albergo et al. (2023a); Albergo & Vanden-Eijnden (2022) with **conditional dependent coupling**. Our approach generates high-resolution images in a coarse-to-fine manner, essentially

addressing the notorious limitations of the multi-stage generation paradigm. Specifically, the proposed method enables efficient and high-performance generation directly in pixel space, thus retaining detailed information without the bottleneck of latent compression. Moreover, by proposing **conditional dependent coupling**, we unify the multi-stage generation process into one coherent framework, ensuring accurate distribution learning at each stage. Notably, the entire multi-stage framework can be parameterized by a single DiT Peebles & Xie (2023b), facilitating knowledge sharing across stages and enabling full end-to-end optimization. We further provide formal proof that the proposed framework significantly reduces the transport cost and inference time. See the Related Works in the Appendix. The source code will be made publicly available upon acceptance.

## 2 PRELIMINARIES

The *stochastic interpolant* Albergo et al. (2023a); Albergo & Vanden-Eijnden (2022) unifies the theory of Ordinary Differential Equations (ODEs) and Stochastic Differential Equations (SDEs). Our method, which was developed based on this theory, is therefore comparable to both the Flow/ODEs and Diffusion/SDEs generation. For notation simplicity and efficient information delivery, in the rest of the paper, we focus on the Flow/ODEs. However, the complete *stochastic interpolant* theory that introduces an additional noise term and recovers Diffusions/SDEs is presented in Appendix §D, where we show that the deterministic interpolant definitions below can be generalized readily.

**Definition 1** (Deterministic interpolant for Flow Matching). Given two probability densities $\rho_0, \rho_1 : \mathbb{R}^d \to \mathbb{R}_{\geq 0}$ and a coupling $\rho(x_0, x_1)$ with marginals $\rho_0$ and $\rho_1$, we define the deterministic interpolant Lipman et al. (2024; 2022):

$$I_t = \alpha_t x_0 + \beta_t x_1, \qquad t \in [0, 1], \tag{1}$$

where $\alpha_t, \beta_t$ are differentiable in $t$ and satisfy the boundary conditions:

$$\alpha_0 = \beta_1 = 1, \qquad \alpha_1 = \beta_0 = 0, \qquad \alpha_t^2 + \beta_t^2 > 0 \ \forall t \in [0, 1].$$

This yields a time–dependent density $\rho_t$ for $I_t$ with $\rho_{t=0} = \rho_0$ and $\rho_{t=1} = \rho_1$.

**Theorem 1** (Transport continuity equation for Flow model). *Let $b_t : \mathbb{R}^d \to \mathbb{R}^d$ denote the* conditional velocity field *Lipman et al. (2024; 2022):*

$$b_t(x) = \mathbb{E}\left[ \dot{I}_t \,\Big|\, I_t = x \right], \tag{2}$$

*where $\dot{f} = \frac{\mathrm{d}f}{\mathrm{d}t}$ and the expectation is over $(x_0, x_1) \sim \rho(x_0, x_1)$. The interpolant density $\rho_t$ solves the transport continuity equation Villani et al. (2008); Villani (2021):*

$$\partial_t \rho_t(x) + \nabla \cdot \big( b_t(x) \, \rho_t(x) \big) = 0. \tag{3}$$

*In practice, we learn a model velocity $\hat{b}_t$ to approximate $b_t$ by minimizing:*

$$L_b(\hat{b}) = \int_0^1 \mathbb{E}\left[ \left| \hat{b}_t(I_t) \right|^2 - 2 \dot{I}_t \cdot \hat{b}_t(I_t) \right] \mathrm{d}t, \tag{4}$$

*which is estimable from samples $(x_0, x_1) \sim \rho(x_0, x_1)$.*

**Corollary 1** (Probability flow ODE). *The transport continuity equation implies that the solution $X_t$ to the probability flow ODE Song et al. (2023); Ma et al. (2024):*

$$\dot{X}_t = b_t(X_t) \tag{5}$$

*matches the interpolant law: if $X_{t=0} \sim \rho_0$, then $X_{t=1} \sim \rho_1$. Hence generative sampling is obtained by drawing $x_0 \sim \rho_0$ and integrating equation 5 from $t=0$ to $t=1$.*

**Remark (Generalization to *stochastic interpolant*).** The Appendix §D introduces the full *stochastic interpolant* $I_t = \alpha_t x_0 + \beta_t x_1 + \gamma_t z$ with $z \sim \mathcal{N}(0, I_d)$, Albergo et al. (2023a); Albergo & Vanden-Eijnden (2022) which recovers diffusion-type models, the score field $s_t(x) = \nabla \log \rho_t(x)$, and the companion objective for $s_t$. All transport-equation statements above remain valid, with equation 2–equation 5 appearing as the $\gamma_t \equiv 0$ case used throughout the main text.

**Definition 2** (Transport cost). Let $X_t(x_0)$ be the solution to the probability flow ODE equation 5 for the initial condition $X_{t=0}(x_0) = x_0 \sim \rho_0$. Then the following inequality holds:

$$\mathbb{E}_{x_0 \sim \rho_0} \left[ |X_{t=1}(x_0) - x_0|^2 \right] \leq \int_0^1 \mathbb{E}[|\dot{I}_t|^2] dt < \infty. \tag{6}$$

Minimizing the left-hand side of equation 6 would achieve the optimal transport in the sense of Benamou-Brenier Benamou & Brenier (2000), and the minimum would give the Wasserstein-2 distance between $\rho_0$ and $\rho_1$ Albergo et al. (2023b).

While for a common *stochastic interpolant*, this transport cost construction was made using the choice $\rho(x_0, x_1) = \rho_0(x_0)\rho_1(x_1)$, so that $x_0$ and $x_1$ were drawn independently from the base and the target, **data-dependent coupling** constructs the joint distribution by $\rho(x_0, x_1) = \rho_0(x_0 \mid x_1)\rho_1(x_1)$ to reduce the transport cost Albergo et al. (2023b), such that:

$$\int_{\mathbb{R}^{3d}} |\dot{I}_t|^2 \rho(x_0, x_1)\rho_z(z)\, dx_0 dx_1 dz \leq \int_{\mathbb{R}^{3d}} |\dot{I}_t|^2 \rho_0(x_0)\rho_1(x_1)\rho_z(z)\, dx_0 dx_1 dz, \tag{7}$$

The bound on the transportation cost in equation 6 is more tightly controlled by the construction of data-dependent couplings Albergo et al. (2023b).

## 3 METHODOLOGY

In this section, we introduce a unified multi-stage generative algorithm designed to efficiently produce high-resolution images without relying on a pretrained VAE, thereby mitigating information and detail loss while enabling end-to-end training. The algorithm proceeds through a sequence of stages, where the resolution is progressively refined in a coarse-to-fine manner. At each stage, the output of the previous stage is further enhanced toward a finer resolution target using *stochastic interpolant* with **conditionally dependent coupling**, thus forming a cascaded architecture and guaranteeing accurate data distribution modeling at each stage. Crucially, all stages are parameterized by a single unified DiT Peebles & Xie (2023a), which facilitates knowledge sharing and supports end-to-end optimization. Under the *stochastic interpolant* framework, the proposed method is naturally compatible with Flow models (ODEs) as well as Diffusion models (SDEs). For clarity, we focus here on deterministic interpolants (Flow/ODEs), while noting that the method extends readily to stochastic interpolants (includes Diffusion/SDEs), see Preliminary for details.

### 3.1 NOTATIONS

**Data and Resolution Hierarchy.** Let the target high-resolution image distribution be defined on $\mathbb{R}^{d_K}$. We define a hierarchy of resolutions, where $k \in \{1, \ldots, K\}$ represents the generation stage: ❶ $x^{(K)} \sim p_{\text{data}}$: A sample from the true high-resolution data distribution. ❷ $x_1^{(k)} \in \mathbb{R}^{d_k}$: A ground-truth image sample at stage $k$, which is generated by downsampling the original high-resolution image.

**Upsampling and Downsampling Operators.** We define operators for changing image resolutions. Let the scaling factor between stage $k-1$ and $k$ be a power of 2: $d_k = 2^{k-1}d_1$, for simplicity.

- *Downsampling Operator $D_k$*: This operator maps an image from resolution $d_k$ to $d_{k-1}$ ($D_k : \mathbb{R}^{d_k} \to \mathbb{R}^{d_{k-1}}$). It operates via neighborhood averaging, where a block of pixels in the higher-resolution image is averaged to a single pixel in the lower-resolution image.

- *Upsampling Operator $U_k$*: This operator maps an image from resolution $d_{k-1}$ to $d_k$ ($U_k : \mathbb{R}^{d_{k-1}} \to \mathbb{R}^{d_k}$). It operates via neighborhood replication, where each pixel from the lower-resolution image is copied to form a block in the higher-resolution image.

These operators are defined such that they are transitive: $D_{k+1 \to k-1} = D_k \circ D_{k+1}$. We can define a composite downsampling operator $D_{K \to k}$ that maps from the highest resolution $d_K$ directly to $d_k$. The ground-truth sample for any stage $k$ is thus defined as $x_1^{(k)} = D_{K \to k}(x^{(K)}) \sim \rho_1^{(k)}(x_1^{(k)}|x^{(K)})$.

### 3.2 MULTI-STAGE FLOW MATCHING MODEL

To implement the unified multi-stage generation process, we define a single unified generative model(Flow/ODEs or Diffusion/SDEs) to transform from the source distribution to the target at each stage $k$ with **conditional dependent coupling**. The $K$ stages are designed as below: ❶ **Stage $k = 1$: Noise-to-Image Generation.** The first stage generates a low-resolution image from a simple source distribution (e.g., Gaussian noise). ❷ **Stage $k > 1$: Image-to-Image Refinement.** In

subsequent stages, the resolution is progressively refined by leveraging **conditionally dependent coupling**, where the probability path is explicitly conditioned on the relationship between the low-resolution source and the high-resolution target images. This cascaded approach breaks down the complex task of high-resolution image generation into a series of more manageable sub-problems. We will elaborate on the **motivation** and **benefits** of the design at the end of this section.

**Stage $k = 1$: Noise-to-Image Generation** The first stage ($k = 1$) of our model is responsible for generating the lowest-resolution image from a simple prior distribution. The model is defined by its source and target distributions and the **conditional dependent coupling** between them. For this foundational stage, the joint distribution is the product of the marginals:

$$\begin{aligned} \rho^{(1)}(x_0^{(1)}, x_1^{(1)}|x^{(K)}) &= \rho_0^{(1)}(x_0^{(1)}|x_1^{(1)})\rho_1^{(1)}(x_1^{(1)}|x^{(K)}) \\ &= \rho_0^{(1)}(x_0^{(1)})\rho_1^{(1)}(x_1^{(1)}|x^{(K)}) \end{aligned} \tag{8}$$

The **target distribution** is the empirical distribution of the lowest-resolution ground-truth images, where $x_1^{(1)} \sim \rho_1^{(1)}(x_1^{(1)}|x^{(K)})$. The **source distribution** is a standard Gaussian distribution denoted as $x_0^{(1)} \sim \rho_0^{(1)}(x_0^{(1)}) = \mathcal{N}(0, \sigma^2 I_{d_1})$, which is independent to $\rho_1^{(1)}(x_1^{(1)}|x^{(K)})$, where can also drive that $\rho_0^{(1)}(x_0^{(1)}|x_1^{(1)}) = \rho_0^{(1)}(x_0^{(1)})$ from equation 8.

**Stage $k > 1$: Image-to-Image Refinement** For all subsequent stages ($k > 1$), the model learns to perform a coarse-to-fine resolution enhancement. The core of these stages is the **conditional dependent coupling** that leverages the structural similarity between the lower-resolution source image and the higher-resolution target image. For each stage $k$, the **target distribution** is the distribution of ground-truth images $x_1^{(k)} \sim \rho_1^{(k)}(x_1^{(k)}|x^{(K)})$ at resolution $d_k$. Instead of drawing from a simple prior, the **source sample** $x_0^{(k)}$ is constructed directly from the corresponding target sample $x_1^{(k)}$: $x_0^{(k)} \sim \rho_0^{(k)}(x_0^{(k)}|x_1^{(k)})$. This is achieved by first applying a deterministic mapping $m_k(x_1^{(k)}) = U_k(D_k(x_1^{(k)}))$ and then adding a small amount of Gaussian noise to ensure the conditional probability is well-defined:

$$x_0^{(k)} = m_k(x_1^{(k)}) + \sigma_k\zeta_k = U_k(D_k(x_1^{(k)})) + \sigma_k\zeta_k \tag{9}$$

where $\zeta_k \sim \mathcal{N}(0, I_{d_k})$ and $\sigma_k$ is a small, stage-dependent noise level, which can be tuned. This construction induces the conditional probability $\rho_0^{(k)}(x_0^{(k)}|x_1^{(k)})$ as so called data-dependent coupling Albergo et al. (2023b). Thus, the joint probability density $\rho^{(k)}(x_0^{(k)}, x_1^{(k)}|x^{(K)})$ is shown as:

$$\rho_0^{(k)}(x_0^{(k)}|x_1^{(k)}) = \mathcal{N}(x_0^{(k)}; m_k(x_1^{(k)}), \sigma_k^2 I_{d_k}) \tag{10}$$

$$\rho^{(k)}(x_0^{(k)}, x_1^{(k)}|x^{(K)}) = \rho_0^{(k)}(x_0^{(k)}|x_1^{(k)})\rho_1^{(k)}(x_1^{(k)}|x^{(K)}) \tag{11}$$

**Model Parameterization** We design a unified DiT model that operates consistently across multiple resolutions by sharing a single set of parameters. To enable the model to recognize and differentiate between these resolutions, we introduce an additional resolution embedding. Specifically, we treat the absolute resolution of the feature map $r_k$ obtained after patch embedding Dosovitskiy et al. (2020) as a conditional signal. This signal is then encoded using sinusoidal positional embeddings Vaswani et al. (2017) $e_k = E(r_k)$ and fused with the timestep embedding by the cross-attention mechanism Vaswani et al. (2017) before being fed into the model. See Appendix §F for details about resolution embedding.

To unify the multi-stage generation, the rescaled timestep strategy Chen et al. (2025); Jin et al. (2024); Yan et al. (2024) is employed to align the multi-stage generation time definition with the standard Flow model Lipman et al. (2022), which generates across the time scope $t \in [0, 1]$. For each stage $k$, the start time point and end time point are defined as $t_0^k$ and $t_1^k$, where $0 \le t_0^k < t_1^k \le 1$, $t_0^k = t_1^{k-1}$, and meet boundary condition $t_0^1 = 0$, $t_1^K = 1$. For simplicity, we construct a linear interpolant by specifying $\alpha_t = 1 - \tau$ and $\beta_t = \tau$ in Definition 1:

$$I_\tau^{(k)} = (1 - \tau) \cdot x_0^{(k)} + \tau \cdot x_1^{(k)}, \tag{12}$$

where $\tau = \frac{t - t_0^k}{t_1^k - t_0^k}$, so that timestep $t \in [0, 1]$ of the entire multi-stage generation process located in each stage $k$: $t \in [t_0^k, t_1^k]$ is rescaled to $\tau \in [0, 1]$ within each stage $k$.

The single unified DiT is parameterized by $b^\theta$ to model all generation stages. For the deterministic interpolant (Flow/ODEs), derived from equation 4, the DiT is optimized by:

$$\mathcal{L}(\theta) = \mathbb{E}_{t\sim[0,1],k,e_k,(x_0^{(k)},x_1^{(k)})\sim\rho^{(k)}(x_0^{(k)},x_1^{(k)}|x^{(K)})} \left[ \left| b^\theta(I_\tau^{(k)},\tau,e_k) \right|^2 - 2\dot{I}_\tau^{(k)} \cdot b^\theta(I_\tau^{(k)},\tau,e_k) \right] \tag{13}$$

where $k = \arg_k\{t \in [t_0^k, t_1^k]\}$, and $\dot{I}_\tau^{(k)} = x_1^{(k)} - x_0^{(k)}$.

## 3.3 CONDITIONAL DEPENDENT COUPLING

For the multi-stage generation model proposed in Sec. 3.2, we employed the **conditional dependent coupling** strategy to unify them. Instead of mapping unstructured noise to an image, our approach maps a structured prior—the upsampled low-resolution image from the previous stage with added stage-dependent noise—to the target high-resolution distribution. The joint probability distribution of the entire multi-stage interpolant with **conditional dependent coupling** is expressed as:

$$\rho(x_1^{(K)}, x_0^{(K)}, x_1^{(K-1)}, \ldots, x_0^{(2)}, x_1^{(1)}, x_0^{(1)}) = \rho(x^{(K)}) \prod_{k=1}^{K} \rho^{(k)}(x_0^{(k)}, x_1^{(k)}|x^{(K)})$$

$$= \rho(x^{(K)}) \prod_{k=1}^{K} \rho_0^{(k)}(x_0^{(k)}|x_1^{(k)})\rho_1^{(k)}(x_1^{(k)}|x^{(K)}), \tag{14}$$

This formulation reveals a key property of our model. Conditioned on the final high-resolution image $x^{(K)}$, the data coupling at any given stage $k$, denoted as $\rho^{(k)}(x_0^{(k)}, x_1^{(k)}|x^{(K)})$, is independent of that at any other stage $k^* \neq k$:

$$\rho^{(k)}(x_0^{(k)}, x_1^{(k)}|x^{(K)}) \perp\!\!\!\perp \rho^{(k^*)}(x_0^{(k^*)}, x_1^{(k^*)}|x^{(K)}), \tag{15}$$

**Sequential dependency as Markov chain** This conditional independence establishes a foundational **Markov chain** Norris (1998) structure across the stages. Consequently, the generative process at inference time proceeds sequentially as follows:

$$\rho(\hat{x}_0^{(1)}, \hat{x}_1^{(1)}, \hat{x}_0^{(2)}, \ldots, \hat{x}_1^{(K-1)}, \hat{x}_0^{(K)}, \hat{x}_1^{(K)}) = \rho^{(1)}(\hat{x}_0^{(1)}, \hat{x}_1^{(1)}) \prod_{k=2}^{K} \rho^{(k)}(\hat{x}_0^{(k)}, \hat{x}_1^{(k)} \mid \hat{x}_1^{(k-1)})$$

$$= \rho_0^{(1)}(\hat{x}_0^{(1)})\rho_1^{(1)}(\hat{x}_1^{(1)} \mid \hat{x}_0^{(1)}) \prod_{k=2}^{K} \rho_1^{(k)}(\hat{x}_1^{(k)} \mid \hat{x}_0^{(k)})\rho_0^{(k)}(\hat{x}_0^{(k)} \mid \hat{x}_1^{(k-1)}), \tag{16}$$

Here, the initial prior $\rho_0^{(1)}(\hat{x}_0^{(1)})$ is a standard normal distribution, $\mathcal{N}(\hat{x}_0^{(1)}; 0, I_{d_1})$. The conditional distributions $\rho_1^{(k)}(\hat{x}_1^{(k)} \mid \hat{x}_0^{(k)})$ for all stages $k$ are parameterized by a single unified generative model $b^\theta$. For the refinement stages ($k \geq 2$), the conditional prior $\rho_0^{(k)}(\hat{x}_0^{(k)} \mid \hat{x}_1^{(k-1)})$ is defined by the deterministic upsampling step with added Gaussian noise:

$$\rho_0^{(k)}(\hat{x}_0^{(k)} \mid \hat{x}_1^{(k-1)}) = \mathcal{N}(\hat{x}_0^{(k)}; U_k(\hat{x}_1^{(k-1)}), \sigma_k^2 I_{d_k}), \tag{17}$$

which is implemented by sampling $\hat{x}_0^{(k)} = U_k(\hat{x}_1^{(k-1)}) + \sigma_k\zeta_k$, akin with the distribution formed by equation 9. See Appendix §E for details of how the joint distribution in equation 14 leads to the Markovian inference process in equation 16. See Appendix §C for the Markov Chain visualization. In contrast to previous autoregressive modeling methods that severely limited by the computational complexity arising from the full-resolution long-history condition Tian et al. (2024b); Ren et al. (2025), and potential complex token/feature pyramid construction Jiao et al. (2025), the proposed generative process begins with a sample from the simple prior: $\hat{x}_0^{(1)} \sim \rho_0^{(1)}(\hat{x}_0^{(1)})$, subsequently, the output of each stage $\hat{x}_1^{(k-1)}$ serves as the basis for the input to the next stage $\hat{x}_0^{(k)}$, forming the sequential dependency characteristic of a Markov chain, as shown in equation 16.

**Reducing transport cost** **Conditional dependent coupling** strategy integrating each stage and thus decreasing transport cost $L = \int_0^1 \mathbb{E}[|\dot{I}_t|^2]dt < \infty$ as defined in Definition 2, see in Theorem 2.

**Theorem 2.** *The transport cost of the naive single-stage model, which generates a high-resolution image $\hat{x}_1 \sim x^{(K)} \in \mathbb{R}^{d_K}$ directly from Gaussian noise $x_0 \sim \mathcal{N}(0, \sigma^2 I_{d_K})$ (denoted as $L_A$), is greater than that of the proposed multi-stage model with conditional dependent coupling strategy, which generates a high-resolution image $\hat{x}_1^{(K)} \sim x^{(K)} \in \mathbb{R}^{d_K}$ from Gaussian noise $x_0^{(1)} \sim \mathcal{N}(0, \sigma^2 I_{d_1})$ through $K$ stages (denoted as $L_B$). Specifically, we have $L_A > L_B$.*

$$L_A = \mathbb{E}[|x_1|^2] + \mathbb{E}[|x_0|^2] \; > \; L_B = \sum_{k=1}^{K} L_k, \tag{18}$$

*where $L_k$ denotes the transport cost at the $k$-th stage. See proof in Appendix §G.1.*

**Accurate distribution learning** In contrast to existing unified multi-stage generation methods Chen et al. (2025); Jin et al. (2024), the proposed method guarantees that the model $b^\theta$ learns the accurate data distribution $\rho_1^{(k)}(x_1^{(k)} \mid x^{(K)})$ at each stage. This, in turn, enables the unified generative model to capture the target high-resolution image distribution $x^{(K)} \sim p_{\text{data}}$ by progressively matching $\rho_1^{(k)}(x_1^{(k)} \mid x^{(K)})$ at every stage $k$. As an example, consider conditional generation: $b^\theta(I_\tau^{(k)}, \tau, e_k)$ models stage-$k$ generation by solving the ODE $\dot{X}_\tau^k = b^\theta(I_\tau^{(k)}, \tau, e_k)$, where $b^\theta(I_\tau^{(k)}, \tau, e_k) = \mathbb{E}_{\mathbf{c}}\left[b^\theta(I_\tau^{(k)}, \tau, e_k, \mathbf{c})\right]$ and $\mathbf{c}$ denotes the conditioning signal (e.g., class label, text prompt). Conditional generation toward conditions $\mathbf{c}_1$ and $\mathbf{c}_2$ within stage $k$ is given by

$$I_{1,\mathbf{c}_1}^{(k)} = I_\tau^{(k)} + \int_\tau^1 b^\theta(I_t^{(k)}, t, e_k, \mathbf{c}_1)\mathrm{d}t, \qquad I_{1,\mathbf{c}_2}^{(k)} = I_\tau^{(k)} + \int_\tau^1 b^\theta(I_t^{(k)}, t, e_k, \mathbf{c}_2)\mathrm{d}t, \tag{19}$$

where $I_\tau^{(k)}$ denotes the current sample at the rescaled timestep $\tau \in [0, 1]$ within stage $k$, and specifically $I_0^{(k)}$ denotes the starting point of stage $k$.

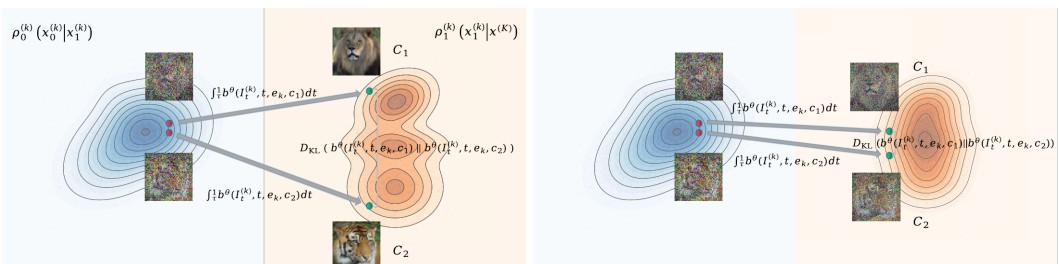

Figure 1: Data Distribution Learning and Transition.

In the initial and intermediate stages, $I_\tau^{(k)}$ approximately follows a standard Gaussian—structureless and semantically vague—due to low resolution and high noise levels, and is therefore not characterizable. Consequently, we assume that the current sample $I_\tau^{(k)}$ used to generate $I_{1,\mathbf{c}_1}^{(k)}$ and $I_{1,\mathbf{c}_2}^{(k)}$ is identical within an initial or intermediate stage $k$; see the red point in Figure 1.

For previous multi-stage methods Chen et al. (2025); Jin et al. (2024) that maintain noise in the target across stage $k < K$, the initial and intermediate stages necessarily target structureless, semantically vague data (close to standard Gaussian noise) due to low resolution and high noise levels. As a result, the model $b^\theta$ cannot learn accurate condition-specific distributions for $\mathbf{c}_1$ and $\mathbf{c}_2$, as indicated by a small divergence $D_{\text{KL}}\left(b^\theta(I_t^{(k)}, t, e_k, \mathbf{c}_1)\big\|b^\theta(I_t^{(k)}, t, e_k, \mathbf{c}_2)\right)$; see the right part of Figure 1.

However, under the proposed training strategy with **conditional dependent coupling**, which treats the accurate data distribution $\rho_1^{(k)}(x_1^{(k)} \mid x^{(K)}) = \mathbb{E}_{\mathbf{c}}\left[\rho_1^{(k)}(x_1^{(k)} \mid x^{(K)}, \mathbf{c})\right]$ as the target at each stage—where $\rho_1^{(k)}(x_1^{(k)} \mid x^{(K)}, \mathbf{c})$ denotes the condition-specific distribution of condition $\mathbf{c}$—the multi-stage generator $b^\theta$ is guaranteed to learn the information and details pertinent to stage $k$ by transforming the source distribution $\rho_0^{(k)}(x_0^{(k)} \mid x_1^{(k)})$ into the accurate target distribution $\rho_1^{(k)}(x_1^{(k)} \mid x^{(K)})$ at stage $k$ for every condition $\mathbf{c}$. This is reflected by an adequate divergence $D_{\text{KL}}\left(b^\theta(I_t^{(k)}, t, e_k, \mathbf{c}_1)\big\|b^\theta(I_t^{(k)}, t, e_k, \mathbf{c}_2)\right)$; see the left part of Figure 1.

### 3.4 Classifier-Free Guidance for Conditional Dependent Coupling

Thanks to the accurate data distribution modeling at each stage $k$, Classifier-Free Guidance (CFG) Ho & Salimans (2022) can be seamlessly integrated with **conditional dependent coupling**. This eliminates the need for a stage-wise CFG schedule Chen et al. (2025); Kynkäänniemi et al. (2024), which otherwise requires carefully assigning the CFG strength at every stage, thereby ensuring both elegance and ease of implementation. In the proposed method, a single CFG strength $S_{\text{cfg}}$ is applied uniformly across all stages $k$. Further details are provided in Appendix §I.

### 3.5 Algorithm Workflows

Here we present the detailed unconditional training and inference procedures for the unified multi-stage generative model $b^{\theta}$ with **conditional dependent coupling** in Appendix §H. The inference steps are described using the forward Euler method Lipman et al. (2024; 2022) for solving the ODE, as a simple example of a numerical solver. Moreover, the detailed training and inference procedures for modeling $b^{\theta}$ in conditional generation with CFG are provided in Appendix §I.

## 4 Theoretical Justification and Key Property

**Mathematical premise and intuition.** Our method adopts a unified multi-scale, coarse-to-fine image generation procedure that aligns with the natural way humans recognize images . Compared to previous coarse-to-fine AR approaches Tian et al. (2024a); Ren et al. (2025), our method is based on a unified and continuous transformation between distributions in a highly controllable manner, which better reflects the smooth nature of image transitions. Compared to existing multi-stage generation methods Ho et al. (2022a), our method models a unified multi-stage generation process with **conditional dependent coupling** and can be parameterized by a single DiT that can be trained from end-to-end. Furthermore, due to the accurate distribution modeling at each stage, our method is guaranteed to learn the target high-quality data distribution by progressively matching the data distribution at each stage. This resolves the generation problem step by step within a unified process.

**Diversity and stochasticity.** In image generation, diversity—typically introduced by the stochasticity of generative models—is primarily expressed in the early stages (initial transition steps for ODE/SDE) Chen et al. (2025); Tian et al. (2024b). For the proposed method, this corresponds to the transition modeled in the initial and intermediate stages. During these stages, key attributes like object presence, layout, and structure are determined. In contrast, the subsequent high-resolution refinement stages require less diversity. By employing the proposed **conditional dependent coupling** strategy, we explicitly decouple the diverse coarse image generation from the fine image refinement, thereby achieving both high performance and efficiency. See Appendix §C for the visualization of the multi-stage generation and the details hierarchy it forms. Specifically, as the stage index $k$ increases, we gradually reduce the stage-dependent noise level $\sigma_k$ in equation 9. This adjustment improves both training and inference efficiency while maintaining strong performance, defined as

$$\sigma_k = \gamma^{-(k-1)}\sigma, \quad \text{where } \gamma \geq 1, \quad 2 \leq k \leq K. \tag{20}$$

We refer to $\gamma$ as the "diminish factor" and further analyze its impact in our experiments.

**Few-step generation and Efficient inference.** Although not originally designed as such, the generative task modeled by the proposed unified multi-stage method with **conditional dependent coupling** is decomposed into subtasks within each generation stage. This decomposition regularizes the overall generation trajectory into multiple straighter sub-trajectories, thereby reducing the number of function evaluations (NFE) required at each stage Frans et al. (2024); Geng et al. (2025b). Moreover, the simulations in the early stages are substantially faster than those in the final stage, which targets the full-resolution image. Quantitatively, this approach leads to a significant reduction in the transport cost, as established in Theorem 2. Consequently, fewer steps are needed to obtain qualified results, and the overall inference time is shortened. Here we summarize as Theorem 3 below.

**Theorem 3.** *The inference time of the naive single-stage model, which generates a high-resolution image $\hat{x}_1 \sim x^{(K)} \in \mathbb{R}^{d_K}$ directly from Gaussian noise $x_0 \sim \mathcal{N}(0, \sigma^2 I_{d_K})$ (denoted as $T_A$), is greater than that of the proposed multi-stage model with a conditional dependent coupling strategy, which generates a high-resolution image $\hat{x}_1^{(K)} \sim x^{(K)} \in \mathbb{R}^{d_K}$ from Gaussian noise*

$x_0^{(1)} \sim \mathcal{N}(0, \sigma^2 I_{d_1})$ *through K stages (denoted as* $T_B$*). Specifically, we have* $T_A > T_B$*.*

$$T_A = \mathbb{E}[\iota(x_0 \to \hat{x}_1)] \; > \; T_B = \sum_{k=1}^{K} \mathbb{E}[\iota(x_0^{(k)} \to \hat{x}_1^{(k)})], \tag{21}$$

*where* $\iota(\cdot)$ *denotes the expected inference time cost of generating the target image from the input sample at each stage. See proof in Appendix §G.2.*

## 5 EXPERIMENT

To comprehensively evaluate the effectiveness of the proposed unified multi-stage generation model with **conditional dependent coupling** (CDC-FM), we conduct extensive experiments.

**Class-conditional image generation.** In Table 1, we compare CDC-FM with various existing methods, including GANs, Autoregressive(AR) models, Diffusion/Flow Matching models, and Multi-stage models in the ImageNet-1k/256 benchmark Krizhevsky et al. (2012), which represent current state-of-the-art generative performance. Furthermore, we also report the comparison result of the ImageNet-1k/512 benchmark Krizhevsky et al. (2012) in Table 2, which shows the robustness of our method in higher resolution. We visualize the class-conditional image generation result of our method in Figure 8, demonstrating high-fidelity, accuracy, and diversity across all the classes.

| | Model/Method | Params | FID↓ | IS↑ | Precision↑ | Recall↑ | NFE | Speed |
|---|---|---|---|---|---|---|---|---|
| GAN | BigGAN Brock et al. (2019) | 112M | 6.95 | 224.5 | 0.89 | 0.38 | 1 | 1.46 |
| | GigaGAN Kang et al. (2023) | 569M | 3.45 | 225.5 | 0.84 | 0.61 | 1 | 1.32 |
| | StyleGAN Karras et al. (2019) | 166M | 2.30 | 265.1 | 0.78 | 0.53 | 1 | 0.96 |
| Diffusion / Flow | ADM Dhariwal & Nichol (2021) | 554M | 10.94 | 101.0 | 0.69 | 0.63 | 250 × 2 | 9.46 |
| | U-ViT Bao et al. (2023) | 287M | 3.40 | 219.9 | 0.83 | 0.52 | - | - |
| | LDM-4-G Rombach et al. (2022) | 400M | 3.60 | 247.7 | - | - | 250 × 2 | 4.18 |
| | DiT-XL/2 Peebles & Xie (2023a) | 675M | 2.27 | 278.2 | 0.83 | 0.57 | 250 × 2 | 3.66 |
| | SiT-XL/2 Ma et al. (2024) | 675M | 2.06 | 270.3 | 0.82 | 0.59 | 250 × 2 | 3.53 |
| | VDM++ Kingma et al. (2023) | 2.0B | 2.12 | 267.7 | - | - | - | - |
| | SiD Hoogeboom et al. (2023) | 2.0B | 2.77 | 211.8 | - | - | 512 × 2 | 10.12 |
| | SiD2 Hoogeboom et al. (2025) | 2.0B | 1.72 | - | - | - | - | - |
| | JetFormer Tschannen et al. (2024) | 2.8B | 6.64 | | 0.69 | 0.56 | | - |
| | REPA Yu et al. (2025b) | 675M | 1.42 | 305.7 | 0.80 | 0.65 | 250 × 2 | 3.56 |
| | LightningDiT Yao et al. (2025) | 675M | 1.35 | 295.3 | 0.79 | 0.65 | 250 × 2 | 3.54 |
| Autoregressive | VQVAE-2 Razavi et al. (2019) | 13.5B | 31.11 | 45.0 | 0.36 | 0.57 | - | - |
| | ViT-VQGAN Yu et al. (2021) | 1.7B | 4.17 | 175.1 | - | - | - | - |
| | VQGAN Esser et al. (2021b) | 1.4B | 15.78 | 74.3 | - | - | 1024 | 12.76 |
| | RQTransformer Lee et al. (2022) | 3.8B | 7.55 | 134.0 | - | - | - | - |
| | LlamaGen-XL Sun et al. (2024) | 775M | 2.62 | 244.1 | 0.80 | 0.57 | - | - |
| | MaskGIT Chang et al. (2022) | 227M | 6.18 | 182.1 | 0.80 | 0.51 | 8 | 0.97 |
| | RCG Li et al. (2023) | 502M | 3.49 | 215.5 | - | - | - | - |
| | MAR-H Li et al. (2024) | 943M | 1.55 | 303.7 | 0.81 | 0.62 | 256 × 2 | 1.23 |
| | FlowAR-L Ren et al. (2024) | 589M | 1.90 | 281.4 | 0.83 | 0.57 | - | - |
| | VAR-d30 Tian et al. (2024b) | 2.0B | 1.92 | 323.1 | 0.82 | 0.59 | 10 × 2 | 1.12 |
| Multi-stage | CDM Ho et al. (2022b) | - | 4.88 | 158.7 | - | - | - | - |
| | PixelFlow Chen et al. (2025) | 677M | 1.98 | 282.1 | 0.81 | 0.60 | - | - |
| | RIN Jabri et al. (2022) | 410M | 3.42 | 182.0 | - | - | - | - |
| | PaGoDA Kim et al. (2024) | 0.9B | 1.56 | 259.6 | - | 0.59 | - | - |
| | FractalMAR-H Li et al. (2025) | 848M | 6.15 | 348.9 | 0.81 | 0.46 | - | - |
| | **CDC-FM (ours)** | 677M | 1.97 | 298.2 | 0.80 | 0.58 | 4 × 4 × 2 | 1.31 |
| | **CDC-FM (ours)** | 677M | 1.87 | 301.8 | 0.82 | 0.60 | 8 × 4 × 2 | 2.66 |
| | **CDC-FM (ours)** | 677M | 1.81 | 304.3 | 0.83 | 0.61 | 16 × 4 × 2 | 5.18 |

Table 1: Comprehensive comparison of generative models on ImageNet-1k/256. Missing values are denoted "-". In the NFE field, "×2" indicates that CFG incurs an NFE of 2 per sampling step.

**Inference Efficiency.** Thanks to the disentangled design of each generation stage in the proposed multi-stage generation model with **conditional dependent coupling**, we explicitly separate coarse image generation from fine image refinement. This disentanglement enables both efficient inference and training while maintaining strong performance. Table 1 reports a comparison of inference time costs between representative methods with CDC-FM under varying numbers of function evaluations (NFE) on the ImageNet-1k/256 benchmark.

Table 2: Comparison on ImageNet-1k/512 conditional generation. Missing values are denoted "-".

| Model/Method | Params | FID↓ | IS↑ |
|---|---|---|---|
| BigGAN Brock et al. (2019) | - | 8.43 | 177.9 |
| ADM Dhariwal & Nichol (2021) | 554M | 7.72 | 172.7 |
| VDM++ Kingma et al. (2023) | 2B | 2.65 | 278.1 |
| DiT-XL/2 Peebles & Xie (2023a) | 675M | 3.04 | 240.8 |
| U-ViT Bao et al. (2023) | 501M | 4.05 | 263.8 |
| SiD2 Hoogeboom et al. (2025) | 2.0B | 2.19 | - |
| MaskGIT Chang et al. (2022) | 227M | 7.32 | 156.0 |
| MAGVIT-v2 Yu et al. (2025a) | 307M | 3.07 | 324.3 |
| MAR-L Li et al. (2024) | 481M | 2.74 | 205.2 |
| VQGAN Esser et al. (2021b) | 1.4B | 26.52 | 66.8 |
| VAR-d36-s Tian et al. (2024b) | 2.0B | 2.63 | 303.2 |
| PaGoDA Kim et al. (2024) | 0.9B | 1.80 | 251.3 |
| **CDC-FM** | 684M | 2.65 | 307.4 |

Furthermore, we measure the inference time of CDC-FM at varying image resolutions: $\{64, 128, 256, 512\}$ on the ImageNet-1k benchmark. Figure 2 showcases the trend that the inference time of the proposed multi-stage generative model with **conditional dependent coupling** is linear to the generated image size, which coincides with Theorem 3 and its proof in Appendix § G.2.

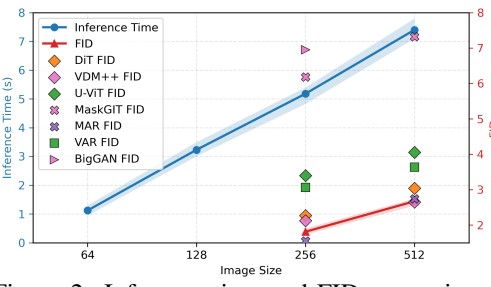

Figure 2: Inference time and FID across image sizes.

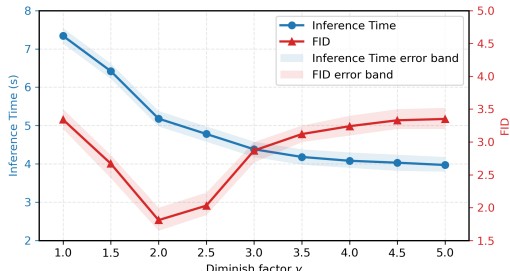

Figure 3: Effect of diminish factor on inference time and FID.

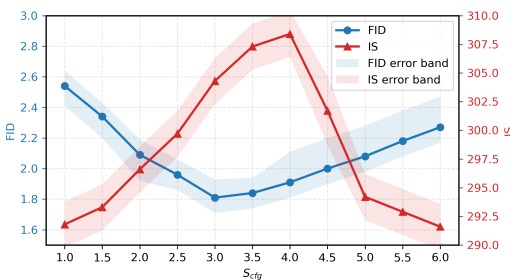

Figure 4: FID and IS under different $S_{\mathrm{cfg}}$.

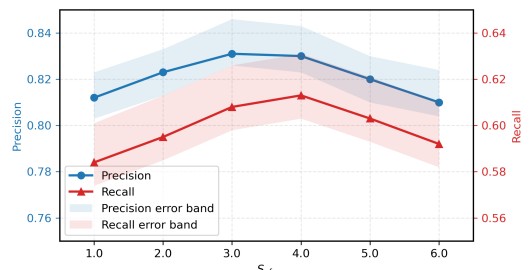

Figure 5: Precision and recall under different $S_{\mathrm{cfg}}$.

**Classifier-free Guidance.** As discussed in Sec. 3.4, we employ a constant CFG strength $S_{\mathrm{cfg}}$ across all stages. This strategy is both simple and elegant to implement, while remaining robust in practice. The performance of CDC-FM on the ImageNet-1k/256 benchmark under varying $S_{\mathrm{cfg}} \in [1.0, 6.0]$ is reported in Figure 4 and Figure 5. To validate the effectiveness of using a constant CFG strength $S_{\mathrm{cfg}}$ across all stages, we report the per-stage FID under varying $S_{\mathrm{cfg}}$. As shown in Figure 7, each stage attains its lowest FID at a similar $S_{\mathrm{cfg}}$ when using **conditional dependent coupling**. This demonstrates that employing a constant CFG strength $S_{\mathrm{cfg}}$ across all stages is both feasible and effective.

**Impact of the Diminish Factor.** We further investigate the effect of the diminish factor $\gamma$ in CDC-FM on the ImageNet-1k/256 benchmark, sweeping $\gamma$ from 1.0 to 5.0 in increments of 0.5 and keeping $\sigma = 1$. The results, summarized in Figure 3, show that the FID initially decreases as $\gamma$ increases, up to a certain tipping point, after which the FID begins to rise. At the same time, the inference time steadily decreases with larger $\gamma$ and eventually plateaus. To test the interaction between $\gamma$ and $\sigma$, we further evaluate the FID under varying $\gamma$ and $\sigma$ in Figure 6.

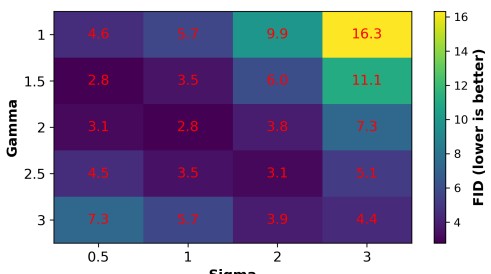

Figure 6: FID under varying $\gamma$ and $\sigma$.

When the stage-$k$ noise level $\sigma_k$ becomes too small (which occurs when $\gamma$ is too large or the base noise level $\sigma$ is too small), each $x_0^{(k)}$ collapses toward a lower-dimensional subset embedded in a higher-dimensional space, causing the induced distribution to concentrate on a lower-dimensional manifold Alberge et al. (2023b). Conversely, when $\sigma_k$ becomes too large (resulting from a too-small $\gamma$ or a too-large $\sigma$), $x_0^{(k)}$ retains insufficient information from the previous stage. Selecting an appropriate $\sigma_k$ for each stage mitigates both extremes. Introducing a small amount of Gaussian noise smooths the base density, ensuring it remains well-defined across the entire ambient space Alberge et al. (2023b), while still preserving adequate information from the preceding stage. This balance enables each stage to refine its input effectively using fewer simulation steps.

**Ablation Study.** To examine the effectiveness of the proposed method and the contribution of each individual component, we conduct a series of ablation studies. These include evaluating the unified model parameterization, the use of interpolants, the choice of resolution hierarchy (i.e., the stage count $K$), and the method by which the resolution embedding is incorporated.

To evaluate whether the unified generative model parameterization enables effective knowledge sharing across stages, we compare the unified DiT (CDC-FM) with stage-specific DiTs on the ImageNet-1k/256 benchmark, reporting FID, model parameters, and training time. For all experiments, we adopt the resolution hierarchy $[32, 64, 128, 256]$ with stage count $K = 4$. For the stage-specific DiTs, we use SiT-B (130M) for the 32, 64, and 128 resolutions, and SiT-L (458M) for the 256 resolution. To ensure a fair comparison, we keep the **conditional dependent coupling** identical to that used in the unified DiT. The results are presented in Table 3. In the unified DiT, the resolution embedding $e_k$ serves as a conditioning signal that informs the model, "you are now operating at resolution $d_k$." This allows all stages $k$ to share the same model parameters, in contrast to stage-specific DiTs, which require separate parameter sets for each resolution. Such unified parameterization encourages efficient knowledge sharing across stages and leads to more compact and effective models.

To further examine the selection of stochastic interpolants, we additionally test alternative interpolants: trigonometric Albergo et al. (2023a), where $\alpha_t = \cos\frac{\pi}{2}\tau$ and $\beta_t = \sin\frac{\pi}{2}\tau$, and score-based diffusion model Song et al. (2021), where $\alpha_t = \sqrt{1 - \tau^2}$ and $\beta_t = \tau$. The results are provided in Table 5.

For the ImageNet-1k/256 experiments, we use the resolution hierarchy $[32, 64, 128, 256]$ with stage count $K = 4$. We also test an alternative hierarchy $[64, 256]$ with stage count $K = 2$. A comparison between the two settings can be found in Table 6.

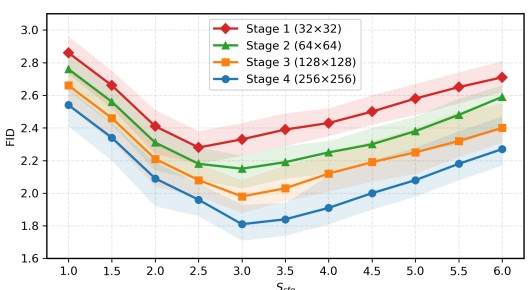

Figure 7: The per-stage FID under varying $S_{\text{cfg}}$.

| Method | FID ↓ | Params (M) ↓ | Training Time ↓ |
|---|---|---|---|
| Unified DiT (CDC-FM) | 1.81 | 677M | 76h |
| Stage-specific DiT | 2.68 | 130M+130M+130M+458M | 78h |

Table 3: Comparison between Unified DiT (CDC-FM) and Stage-specific DiT (implemented by cascade SiT Ma et al. (2024)) on ImageNet-1k/256.

| Model Variant | FID ↓ |
|---|---|
| With Cross-Attention | 1.81 |
| Without Cross-Attention | 1.84 |

Table 4: Ablation on cross-attention fusion mechanism of resolution embedding $e_k$.

| Interpolant | Definition | FID ↓ |
|---|---|---|
| Linear Lipman et al. (2022) | $\alpha_t = 1 - \tau;\ \beta_t = \tau$ | 1.81 |
| Trigonometric Albergo et al. (2023a) | $\alpha_t = \cos\frac{\pi}{2}\tau;\ \beta_t = \sin\frac{\pi}{2}\tau$ | 2.11 |
| Score-based Diffusion Song et al. (2021) | $\alpha_t = \sqrt{1 - \tau^2};\ \beta_t = \tau$ | 2.03 |

Table 5: FID comparison using alternative interpolants. Trigonometric interpolant follows Albergo et al. (2023a); score-based diffusion interpolant follows Song et al. (2021).

| Stages $K$ | Resolution Path | FID ↓ |
|---|---|---|
| 4 | [32, 64, 128, 256] | 1.81 |
| 2 | [64, 256] | 1.97 |

Table 6: Comparison between the standard 4-stage hierarchy and the alternative 2-stage hierarchy on ImageNet-1k/256.

Here, we additionally evaluate a variant that drops cross-attention and fuses $e_k$ with the timestep embedding via direct addition. The results of this comparison are shown in Table 4.

## 6 CONCLUSION

We develop a unified multi-stage generation framework based on *stochastic interpolant* with **conditional dependent coupling**, which enables accurate data distribution learning while operating efficiently in pixel space. Moreover, the entire process can be parameterized by a single DiT, thereby achieving end-to-end optimization and facilitating knowledge sharing across stages. We further provide formal proof that the proposed framework significantly reduces the transport cost and inference time. Diverse experiments demonstrate that our method achieves SOTA performance across metrics and is capable of being implemented in higher resolution.

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

APPENDIX CONTENTS

## A    RELATED WORKS

**Latent Space Generation.** Variational Autoencoders (VAEs) enable generative models to perform Flow/Diffusion generation, as well as autoregressive generation, within a lower-dimensional latent space Rombach et al. (2022); Peebles & Xie (2023a); Ma et al. (2024); Razavi et al. (2019); Esser et al. (2021b); Yu et al. (2021); Lee et al. (2022); Sun et al. (2024); Ren et al. (2024); Tian et al. (2024b); Jabri et al. (2022). This leads to more efficient training and inference. However, the

compact representations produced by the encoder and the subsequent decoding process inevitably introduce a degree of information and detail loss Li et al. (2024); Hoogeboom et al. (2025; 2023). Despite this limitation, VAEs remain a crucial component in these generative models. Moreover, they typically cannot be trained jointly with the generative model in a fully end-to-end manner Chen et al. (2025).

**Pixel Space Generation.** Although directly implementing the generation process avoids the need for VAEs and facilitates the preservation of fine-grained details, it is highly inefficient for both training and inference Rombach et al. (2022); Hoogeboom et al. (2025; 2023); Bao et al. (2023). Furthermore, due to the high dimensionality and complexity of the data distribution, learning to generate samples that faithfully capture detailed information from the target distribution becomes particularly challenging, especially in the absence of VAEs to extract meaningful latent features Chen et al. (2025); Jin et al. (2024); Ren et al. (2024); Tian et al. (2024b); Jabri et al. (2022); Sun et al. (2024).

**Multi-stage Generation.** Multi-stage generation methods advance sample synthesis by decomposing the process into a sequence of stages, where early stages operate in a lower-dimensional space to improve efficiency Chen et al. (2025); Jin et al. (2024); Tian et al. (2024b); Jiao et al. (2025); Ren et al. (2025); Ho et al. (2022b); Jabri et al. (2022). In addition, the overall generative task is divided into subtasks at each stage, enabling the model to gradually capture the complexity of the target data distribution Tian et al. (2024b); Jiao et al. (2025); Ren et al. (2025); Ho et al. (2022b). However, the use of disentangled stage designs hinders unified model parameterization and end-to-end optimization Chen et al. (2025); Jin et al. (2024). Furthermore, the decoupling inherent in multi-stage frameworks may lead to inaccurate modeling of the data distribution, thereby introducing errors that can accumulate across stages Chen et al. (2025); Jin et al. (2024).

## B  QUALITATIVE RESULTS

We visualize the class-conditional image generation result of our method in Figure 8, demonstrating high-fidelity, accuracy, and diversity across all the classes.

## C  VISUALIZATION OF UNIFIED MODEL WITH CONDITIONAL DEPENDENT COUPLING

In this section, we visualize representative examples of the output produced at each stage of the proposed method, together with the details added at each stage. The results shown in Figure 9 illustrate that the proposed framework indeed forms a Markov chain: conditioned on the output of stage $k - 1$, the behavior of stage $k$ is independent of stage $k - 2$. For a detailed explanation of how the joint distribution in equation 14 induces the Markovian inference process in equation 16, please refer to Appendix §E.

We also observe that the magnitude of the added details decreases as the stage index $k$ increases. This indicates that the method decomposes the generation process into multiple stages in a coarse-to-fine manner: earlier stages focus on generating diverse coarse structures, while later stages progressively refine fine-grained details. Such decoupling enables both strong generative performance and improved sampling efficiency.

## D  STOCHASTIC INTERPOLANT

The stochastic interpolant unifies the theory of Ordinary Differential Equations (ODEs) and Stochastic Differential Equations (SDEs).

**Definition 3** (Stochastic Interpolant). Albergo et al. (2023a;b); Albergo & Vanden-Eijnden (2022) Given two probability density functions $\rho_0, \rho_1 : \mathbb{R}^d \to \mathbb{R}_{\geq 0}$, a stochastic interpolant between $\rho_0$ and $\rho_1$ is a stochastic process $I_t$ defined as

$$I_t = \alpha_t x_0 + \beta_t x_1 + \gamma_t z, \qquad t \in [0, 1] \tag{22}$$

where $\alpha_t$, $\beta_t$, and $\gamma_t^2$ are differentiable functions of time satisfying the boundary conditions

$$\alpha_0 = \beta_1 = 1, \quad \alpha_1 = \beta_0 = \gamma_0 = \gamma_1 = 0, \quad \text{and} \quad \alpha_t^2 + \beta_t^2 + \gamma_t^2 > 0 \quad \forall t \in [0, 1].$$

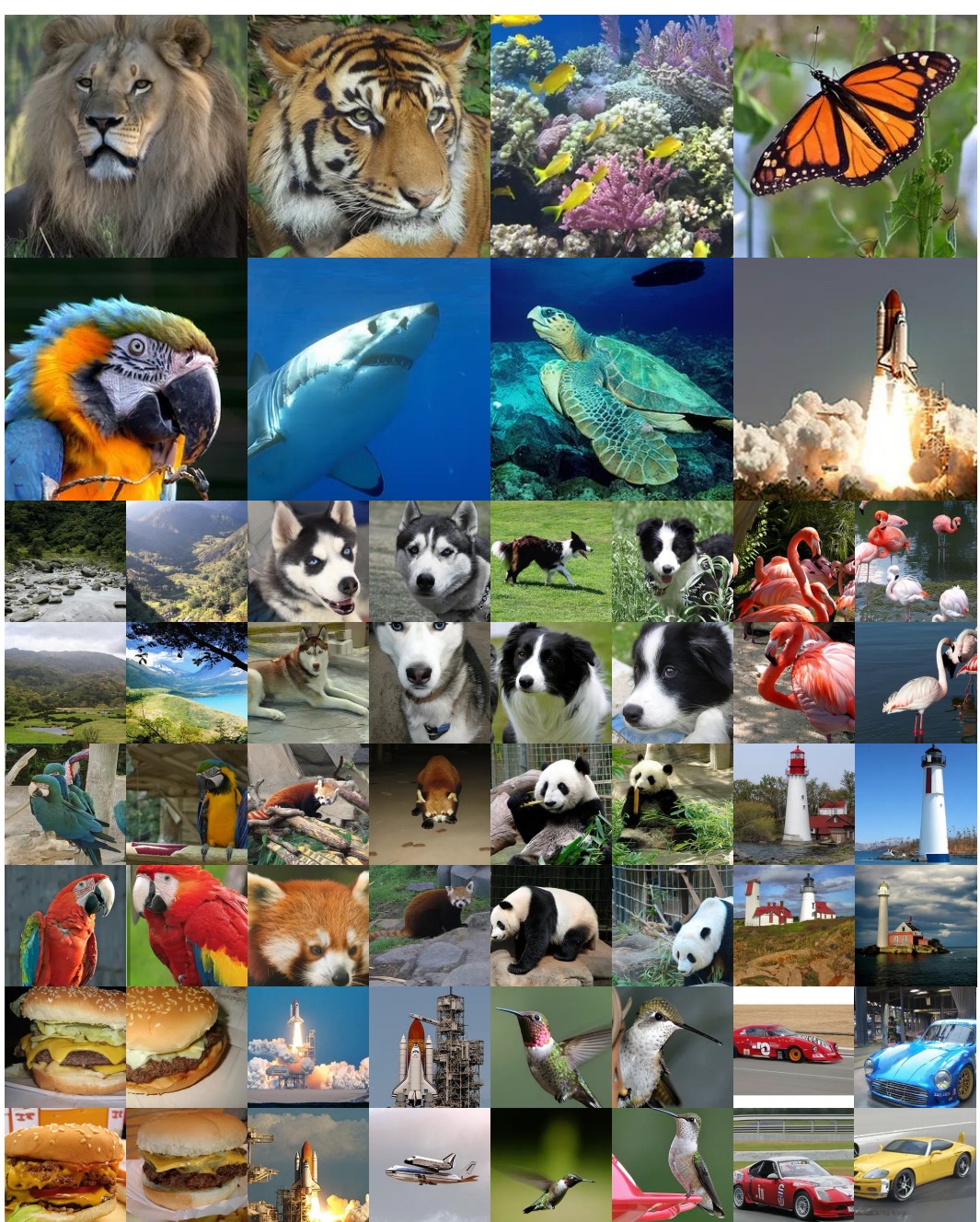

Figure 8: ImageNet-1k/256 class-conditional generation results.

The pair $(x_0, x_1)$ is drawn from a joint probability density $\rho(x_0, x_1)$ with finite second moments, whose marginals are $\rho_0$ and $\rho_1$:

$$\int_{\mathbb{R}^d} \rho(x_0, x_1) dx_1 = \rho_0(x_0), \tag{23}$$

$$\int_{\mathbb{R}^d} \rho(x_0, x_1) dx_0 = \rho_1(x_1), \tag{24}$$

and $z \sim \mathcal{N}(0, \text{Id})$ a Gaussian random variable independent of $(x_0, x_1)$, i.e., $z \perp (x_0, x_1)$.

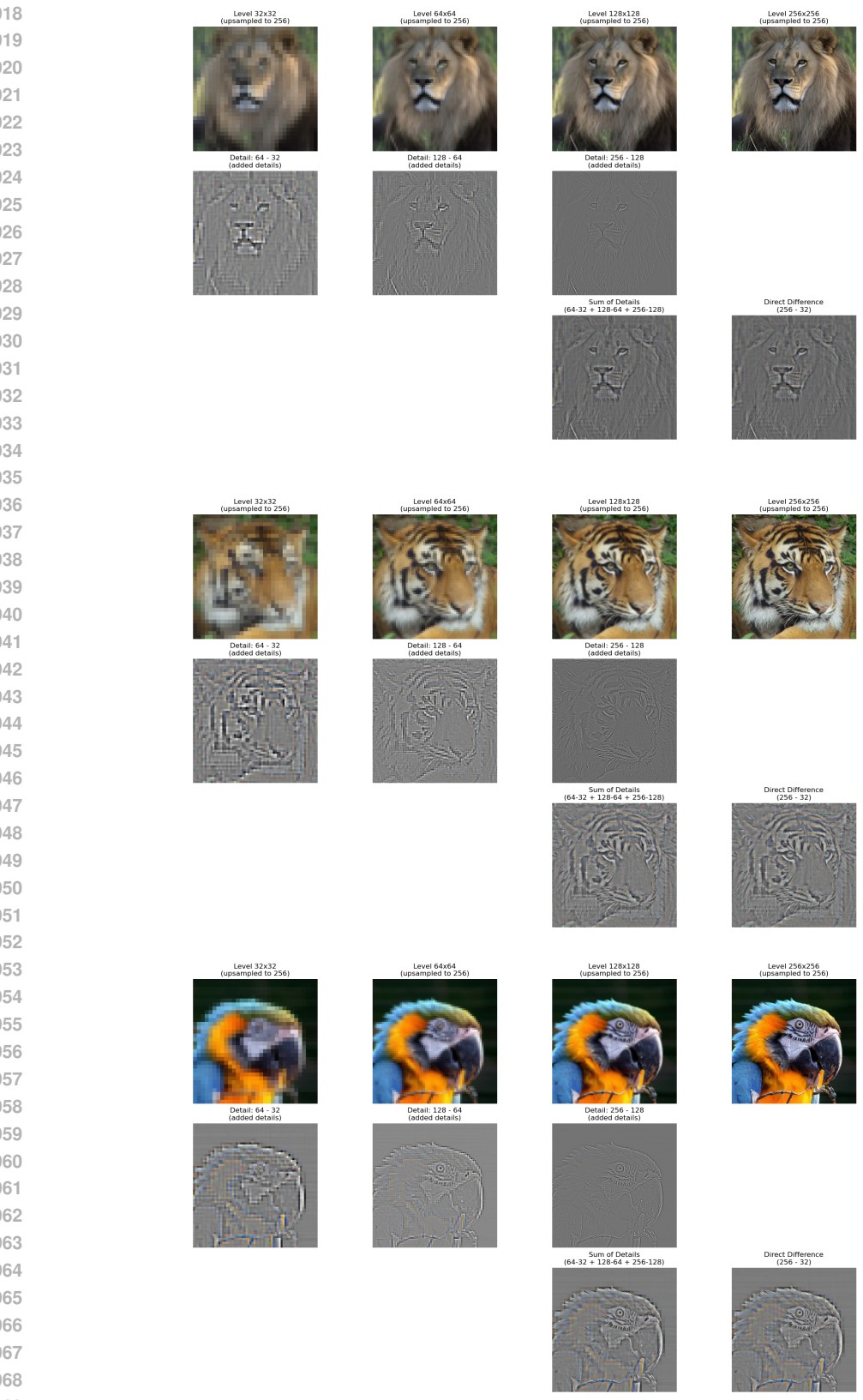

Figure 9: Visualization of each stage in CDC-FM, including the output at every stage, the details added at each stage, and the corresponding hierarchical structure of those details.

The stochastic interpolant framework uses information about the process $I_t$ to derive either an ODE or an SDE. The solutions $X_t$ to these equations are designed to push the initial law $\rho_0$ onto the law of the interpolant $I_t$ for all times $t \in [0,1]$. Consequently, the process $I_t$ satisfies $I_{t=0} = x_0 \sim \rho_0(x_0)$ and $I_{t=1} = x_1 \sim \rho_1(x_1)$. This property allows for generative modeling: by drawing samples $x_0 \sim \rho_0(x_0)$ and using them as initial conditions $X_{t=0} = x_0$, one can generate samples $X_{t=1} \sim \rho_1(x_1)$ via numerical integration of the corresponding ODE or SDE.

**Theorem 4** (Transport equation). *Let the time-dependent density of the stochastic interpolant $I_t$ be $\rho_t(x)$. We define the velocity field $b_t(x)$ and the score field $s_t(x)$ as:*

$$b_t(x) = \mathbb{E}[\dot{I}_t \mid I_t = x] \tag{25}$$

$$s_t(x) = \nabla \log \rho_t(x) \tag{26}$$

*where the dot denotes the time-derivative: $\dot{f} = \frac{\mathrm{d}f}{\mathrm{d}t}$ and the expectation is over $\rho(x_0, x_1)$ and $z$ conditional on $I_t = x$. The probability density $\rho_t(x)$ satisfies the boundary conditions $\rho_{t=0}(x) = \rho_0(x)$ and $\rho_{t=1}(x) = \rho_1(x)$, and solves the transport equation:*

$$\partial_t \rho_t(x) + \nabla \cdot (b_t(x)\rho_t(x)) = 0. \tag{27}$$

*Moreover, for every $t$ such that $\gamma_t \neq 0$, the score is given by:*

$$s_t(x) = -\gamma_t^{-1}\mathbb{E}(z \mid I_t = x). \tag{28}$$

*Finally, the fields $b_t$ and $s_t$ are the unique minimizers of the respective objective functions:*

$$L_b(\hat{b}) = \int_0^1 \mathbb{E}\left[\left|\hat{b}_t(I_t)\right|^2 - 2\dot{I}_t \cdot \hat{b}_t(I_t)\right] dt, \tag{29}$$

$$L_s(\hat{s}) = \int_0^1 \mathbb{E}\left[|\hat{s}_t(I_t)|^2 + 2\gamma_t^{-1}z \cdot \hat{s}_t(I_t)\right] dt, \tag{30}$$

*where $\mathbb{E}$ denotes an expectation over $(x_0, x_1) \sim \rho(x_0, x_1)$ and $z \sim \mathcal{N}(0, \mathrm{Id})$.*

The objective functions equation 29 and equation 30 can be readily estimated in practice from samples $(x_0, x_1) \sim \rho(x_0, x_1)$ and $z \sim \mathcal{N}(0, 1)$, which will enable us to learn approximations for use in a generative model. The transport equation 27 can be used to derive generative models, as we now show.

**Corollary 2** (Probability flow (ODE) and diffusions (SDE)). *The transport equation 27 implies that the density of the solutions $X_t$ to the probability flow ODE matches the interpolant density $\rho_t$. The solutions to the probability flow equation:*

$$\dot{X}_t = b_t(X_t) \tag{31}$$

*satisfy the properties that*

$$X_{t=1} \sim \rho_1(x_1) \quad \textit{if} \quad X_{t=0} \sim \rho_0(x_0), \tag{32}$$

$$X_{t=0} \sim \rho_0(x_0) \quad \textit{if} \quad X_{t=1} \sim \rho_1(x_1). \tag{33}$$

*In addition, for any choice of a time-dependent diffusion coefficient $\epsilon_t \geq 0$, solutions to the forward SDE*

$$dX_t^F = \left(b_t(X_t^F) + \epsilon_t s_t(X_t^F)\right) dt + \sqrt{2\epsilon_t}dW_t, \tag{34}$$

*satisfy the property that*

$$X_{t=1}^F \sim \rho_1(x_1) \quad \textit{if} \quad X_{t=0}^F \sim \rho_0(x_0). \tag{35}$$

*And solutions to the backward SDE*

$$dX_t^R = \left(b_t(X_t^R) - \epsilon_t s_t(X_t^R)\right) dt + \sqrt{2\epsilon_t}dW_t, \tag{36}$$

*satisfy the property that*

$$X_{t=0}^R \sim \rho_0(x_0) \quad \textit{if} \quad X_{t=1}^R \sim \rho_1(x_1). \tag{37}$$

Both deterministic and stochastic generative models were derived within the stochastic interpolant framework.

# E JUSTIFICATION OF CONDITIONAL DEPENDENT COUPLING AND MARKOVIAN INFERENCE

In this appendix, we make explicit how the joint distribution in equation 14 leads to the Markovian inference process in equation 16. We also clarify the equivalence between the training-time coupling and the inference-time generative prior.

## E.1 FROM FULL CONDITIONING TO MARKOVIAN INFERENCE

During training, the couplings $\rho^{(k)}(x_0^{(k)}, x_1^{(k)} \mid x^{(K)})$ are defined with explicit conditioning on the full-resolution image $x^{(K)}$ through $x_1^{(k)} \sim \rho_1^{(k)}(x_1^{(k)} | x^{(K)})$, which is implemented by $x_1^{(k)} = D_{K \to k}(x^{(K)})$, and equation 10, which is implemented by equation 9. At inference time, however, we no longer have access to $x^{(K)}$ and instead conduct a *generative process* that reverses the data-construction direction, analogous to forward/backward procedures in flow or diffusion models:

- In diffusion, one constructs noisy images by repeatedly *adding* noise to a clean image during training, and then learns to *remove* noise step by step at inference.
- In flow matching, one defines an interpolant from source to target and trains a vector field so that the backward integration maps from source to data.

In our setting, the "forward" direction (data-pair construction for training) at stage $k$ is

$$x_1^{(k)} \sim \rho_1^{(k)}(x_1^{(k)} \mid x^{(K)}), \quad x_0^{(k)} \sim \rho_0^{(k)}(x_0^{(k)} \mid x_1^{(k)}),$$

and the "backward" (inference) direction uses the reversed conditional

$$\hat{x}_1^{(k)} \sim \rho_1^{(k)}(\hat{x}_1^{(k)} \mid \hat{x}_0^{(k)}),$$

together with the prior

$$\hat{x}_0^{(k)} \sim \rho_0^{(k)}(\hat{x}_0^{(k)} \mid \hat{x}_1^{(k-1)}),$$

for $k \geq 2$, and $\hat{x}_0^{(1)} \sim \rho_0^{(1)}(\hat{x}_0^{(1)})$ at the first stage.

By construction, the inference process is *sequential*:

1. Sample the initial latent from the simple prior
$$\hat{x}_0^{(1)} \sim \rho_0^{(1)}(\hat{x}_0^{(1)}) = \mathcal{N}(\hat{x}_0^{(1)}; 0, I_{d_1}).$$

2. For stage $k = 1$, sample
$$\hat{x}_1^{(1)} \sim \rho_1^{(1)}(\hat{x}_1^{(1)} \mid \hat{x}_0^{(1)}).$$

3. For each refinement stage $k \geq 2$:
$$\hat{x}_0^{(k)} \sim \rho_0^{(k)}(\hat{x}_0^{(k)} \mid \hat{x}_1^{(k-1)}), \qquad \hat{x}_1^{(k)} \sim \rho_1^{(k)}(\hat{x}_1^{(k)} \mid \hat{x}_0^{(k)}).$$

Since at step $k \geq 2$ the pair $(\hat{x}_0^{(k)}, \hat{x}_1^{(k)})$ depends only on $\hat{x}_1^{(k-1)}$, we obtain the Markov factorization:

$$
\begin{aligned}
&\rho(\hat{x}_0^{(1)}, \hat{x}_1^{(1)}, \hat{x}_0^{(2)}, \ldots, \hat{x}_1^{(K-1)}, \hat{x}_0^{(K)}, \hat{x}_1^{(K)}) \\
&= \rho^{(1)}(\hat{x}_0^{(1)}, \hat{x}_1^{(1)}) \prod_{k=2}^{K} \rho^{(k)}(\hat{x}_0^{(k)}, \hat{x}_1^{(k)} \mid \hat{x}_1^{(k-1)}) \\
&= \rho_0^{(1)}(\hat{x}_0^{(1)}) \rho_1^{(1)}(\hat{x}_1^{(1)} \mid \hat{x}_0^{(1)}) \prod_{k=2}^{K} \rho_1^{(k)}(\hat{x}_1^{(k)} \mid \hat{x}_0^{(k)}) \, \rho_0^{(k)}(\hat{x}_0^{(k)} \mid \hat{x}_1^{(k-1)}),
\end{aligned}
\tag{38}
$$

which is exactly equation 16.

Thus, equation 16 is the natural *reverse* generative process induced by the training-time couplings:

- $\rho_0^{(k)}(x_0^{(k)} \mid x_1^{(k)})$ is used in the forward (data-pair construction) direction.
- $\rho_1^{(k)}(\hat{x}_1^{(k)} \mid \hat{x}_0^{(k)})$ is its reverse counterpart at inference.

The unified DiT $b^\theta$ parameterizes $\rho_1^{(k)}(\hat{x}_1^{(k)} \mid \hat{x}_0^{(k)})$ for all $k$, analogous to how a single score network or vector field is shared across timesteps in diffusion and flow matching models.

## E.2 EQUIVALENCE OF TRAINING AND INFERENCE PRIORS AT REFINEMENT STAGES

We now show that the refinement-stage prior equation 17 used at inference, which is implemented by:

$$\hat{x}_0^{(k)} = U_k(\hat{x}_1^{(k-1)}) + \sigma_k \zeta_k$$

conducts the same conditional distribution as $\rho_0^{(k)}(x_0^{(k)} \mid x_1^{(k)})$ in equation 10, which is implemented by equation 9, once we account for the downsampling hierarchy.

First, recall the *transitivity* of the downsampling operators defined in Sec. 3.1:

$$D_{k+1 \to k-1} = D_k \circ D_{k+1}, \qquad D_{K \to k-1} = D_k \circ D_{K \to k}.$$

By definition of the stage-$k$ ground-truths,

$$x_1^{(k)} = D_{K \to k}(x^{(K)}) \quad \Rightarrow \quad D_k(x_1^{(k)}) = D_k(D_{K \to k}(x^{(K)})) = D_{K \to k-1}(x^{(K)}) = x_1^{(k-1)}.$$

Hence, for the *data-construction process* used in training,

$$m_k(x_1^{(k)}) = U_k(D_k(x_1^{(k)})) = U_k(x_1^{(k-1)}).$$

Plugging this into equation 9 gives

$$x_0^{(k)} = U_k(x_1^{(k-1)}) + \sigma_k \zeta_k, \quad \zeta_k \sim \mathcal{N}(0, I_{d_k}),$$

so that

$$x_0^{(k)} \sim \mathcal{N}(x_0^{(k)}; U_k(x_1^{(k-1)}), \sigma_k^2 I_{d_k}).$$

At inference time, we define the refinement prior by:

$$\hat{x}_0^{(k)} = U_k(\hat{x}_1^{(k-1)}) + \sigma_k \zeta_k, \quad \zeta_k \sim \mathcal{N}(0, I_{d_k}),$$

which induces to equation 17:

$$\rho_0^{(k)}(\hat{x}_0^{(k)} \mid \hat{x}_1^{(k-1)}) = \mathcal{N}(\hat{x}_0^{(k)}; U_k(\hat{x}_1^{(k-1)}), \sigma_k^2 I_{d_k}).$$

Comparing the two expressions, we see that:

- During *training*, when we construct pairs $(x_0^{(k)}, x_1^{(k)})$ from data, the conditional distribution
$$x_0^{(k)} \mid x_1^{(k)} \sim \mathcal{N}(U_k(x_1^{(k-1)}), \sigma_k^2 I_{d_k}),$$
with $x_1^{(k-1)} = D_k(x_1^{(k)})$.

- During *inference*, for any given input $\hat{x}_1^{(k-1)}$ (which is the output of the previous stage), we use
$$\hat{x}_0^{(k)} \mid \hat{x}_1^{(k-1)} \sim \mathcal{N}(U_k(\hat{x}_1^{(k-1)}), \sigma_k^2 I_{d_k}).$$

Therefore, the distribution in equation 17 implemented by

$$\hat{x}_0^{(k)} = U_k(\hat{x}_1^{(k-1)}) + \sigma_k \zeta_k$$

is *identical in form* to the training conditional distribution in equation 10

$$x_0^{(k)} \sim \rho_0^{(k)}(x_0^{(k)} \mid x_1^{(k)})$$

once we use the relation

$$x_1^{(k-1)} = D_k(x_1^{(k)}).$$

In other words, the refinement prior at stage $k$ during inference is exactly the same Gaussian conditional as the one used to generate $x_0^{(k)}$ from $x_1^{(k)}$ during data-pair construction, with $\hat{x}_1^{(k-1)}$ playing the role of $x_1^{(k-1)}$. This is precisely the sense: $\rho_1^{(k)}(\hat{x}_1^{(k)} \mid \hat{x}_0^{(k)})$ is the *reverse* counterpart of $\rho_0^{(k)}(x_0^{(k)} \mid x_1^{(k)})$, and why the manually constructed joint distribution in equation 14 leads to the Markovian generative process in equation 16.

# F    DETAILED IMPLEMENTATION OF RESOLUTION EMBEDDING

$e_k$ is fused with the timestep embedding by the cross-attention mechanism before being fed into the model. The detailed fuse procedure is illustrated below. For each stage $k$ and rescaled timestep $\tau \in [0, 1]$, We follow the standard DiT sinusoidal embedding followed by an MLP:

$$\gamma_t = \Gamma(\tau) \in \mathbb{R}^{d_{\text{model}}}. \tag{39}$$

Given the absolute resolution $r_k$ of stage $k$, we construct the resolution embedding using an analogous sinusoidal embedding followed by an MLP:

$$e_k = E(r_k) \in \mathbb{R}^{d_{\text{model}}}. \tag{40}$$

Both $\gamma_t$ and $e_k$ are 1D vectors in the same model space $\mathbb{R}^{d_{\text{model}}}$.

We implement the fusion by cross-attention, where the timestep embedding $\gamma_t$ is used as the query and the resolution embedding $e_k$ is used as the key/value. Let $d_h$ be the head dimension. We first define linear projections

$$q_t = W_q \gamma_t \in \mathbb{R}^{d_h}, \quad k_k = W_k e_k \in \mathbb{R}^{d_h}, \quad v_k = W_v e_k \in \mathbb{R}^{d_h}, \tag{41}$$

where $W_q, W_k, W_v \in \mathbb{R}^{d_h \times d_{\text{model}}}$.

The cross-attention between the timestep and the resolution embeddings is then

$$\alpha_{k,t} = \text{softmax}\left(\frac{q_t^\top k_k}{\sqrt{d_h}}\right) \in \mathbb{R}, \tag{42}$$

$$z_{k,t} = W_o\big(\alpha_{k,t} \, v_k\big) \in \mathbb{R}^{d_{\text{model}}}, \quad W_o \in \mathbb{R}^{d_{\text{model}} \times d_h}. \tag{43}$$

We then obtain a fused conditioning vector via a residual connection:

$$c_{k,t} = \gamma_t + z_{k,t} \in \mathbb{R}^{d_{\text{model}}}. \tag{44}$$

The vector $c_{k,t}$ can be interpreted as a timestep embedding conditioned on the current stage resolution, and is used as the global conditioning vector for the DiT.

Let $X_0^{(k)} \in \mathbb{R}^{N_k \times d_{\text{model}}}$ denote the sequence of patch tokens at stage $k$ after patch embedding of $I_\tau^{(k)}$. The unified DiT is conditioned on the fused vector $c_{k,t}$. We use $c_{k,t}$ as a global conditioning signal for all DiT blocks via AdaLN modulation Peebles & Xie (2023b), following DiT Peebles & Xie (2023b), SiT Ma et al. (2024) and PixelFlow Chen et al. (2025):

$$X^{\ell+1} = X^\ell + F^{(\ell)}\big(\text{LN}(X^\ell); c_{k,t}\big), \quad \ell = 1, \dots, L, \tag{45}$$

where each block $F^{(\ell)}$ is a DiT block whose parameters are produced from $c_{k,t}$ by AdaLN.

Conceptually, the same DiT parameters are reused for all stages, while the pair $(\tau, r_k)$ is compressed into the single fused conditioner $c_{k,t}$, which tells the model both *where in time* and *at which resolution* it is operating.

# G    PROOF

## G.1    PROOF OF THEOREM 2

We provide a formal proof that a cascaded image generation model using Flow Matching with **conditional dependent coupling** has a significantly lower transportation cost than a direct, single-stage model that generates a high-resolution image from Gaussian noise, as shown in Theorem 2. The proof is based on an orthogonal decomposition of the image signal energy, which reveals that the cost difference is primarily driven by the dimensionality of the initial noise space.

### G.1.1    PROBLEM FORMULATION AND DEFINITIONS

Our goal is to prove that the transportation cost of a direct Flow Matching model ($L_A$) is greater than the total transport cost of a multi-stage cascaded model with conditional dependent couplings ($L_B$).

**Proposition 1** (Transportation Cost Bound). *From Definition 2, the transportation cost of a probability flow is bounded by the integral of the expected squared norm of the interpolant's time derivative. For an interpolant path from a source distribution $\rho_0(x_0)$ to a target distribution $\rho_1(x_1)$, we denote this bound as $L$:*

$$Cost = \mathbb{E}_{x_0 \sim \rho_0} \left[ |X_{t=1}(x_0) - x_0|^2 \right] \leq \int_0^1 \mathbb{E} \left[ |\dot{I}_t|^2 \right] dt := L. \tag{46}$$

*For simplicity in the proof, we analyze the case where the interpolant is $I_t = (1 - t)x_0 + tx_1$, such that its derivative is $\dot{I}_t = x_1 - x_0$. The logic holds for general interpolants. In this case, the cost bound reduces to Lipman et al. (2024); Albergo et al. (2023b):*

$$L = \mathbb{E}[|x_1 - x_0|^2]. \tag{47}$$

Model A: Direct Generation This model generates a high-resolution image $x^{(K)} \in \mathbb{R}^{d_K}$ in a single stage.

- **Source Distribution** $\rho_0(x_0)$: $x_0 \sim \mathcal{N}(0, \sigma^2 I_{d_K})$.

- **Target Distribution** $\rho_1(x_1)$: The high-resolution data distribution, $x_1 \sim x^{(K)}$.

- **Cost** ($L_A$): The transport cost from $x_0$ to $x_1$.

Model B: Cascaded Generation This model generates the image in $K$ stages, from resolution $d_1$ to $d_K$.

- **Stage 1:** Generates a low-resolution image $x_1^{(1)} \in \mathbb{R}^{d_1}$.

    - Source: $x_0^{(1)} \sim \mathcal{N}(0, \sigma^2 I_{d_1})$.
    - Target: $x_1^{(1)} \sim \rho_1^{(1)}(x_1^{(1)}|x^{(K)})$.
    - Cost: $L_1$.

- **Stages** $k \in [2, K]$**:** Refines a coarse image to a higher resolution image $x_1^{(k)} \in \mathbb{R}^{d_k}$.

    - Target: $x_1^{(k)} \sim \rho_1^{(k)}(x_1^{(k)}|x^{(K)})$.
    - Source: $x_0^{(k)} \sim \rho_0^{(k)}(x_0^{(k)}|x_1^{(k)})$ is defined by a deterministic **conditional dependent coupling** with a small stage-dependent Gaussian noise: $x_0^{(k)} = U_k(D_k(x_1^{(k)})) + \sigma_k \zeta_k$, as shown in equation 9.
    - Cost: $L_k$.

- **Total Cost** ($L_B$): The sum of the costs of all stages, $L_B = \sum_{k=1}^{K} L_k$.

We seek to prove that $L_A > L_B$.

G.1.2   ORTHOGONAL DECOMPOSITION OF IMAGE ENERGY

We formalize the concept that an image can be represented as a base layer plus a sum of details.

**Definition 4** (Multiresolution Projections). Let the vector space be that of the highest resolution, $\mathbb{R}^{d_K}$. We define an embedding operator $U_{k \to K} : \mathbb{R}^{d_k} \to \mathbb{R}^{d_K}$ that upscales a level-$k$ image to the full resolution space. We then define a projection operator $D_{K \to k} : \mathbb{R}^{d_K} \to \mathbb{R}^{d_k}$ that projects a high-resolution image onto the subspace of images with resolution $k$. These two definitions are inherited from the previous section. For an image $x \in \mathbb{R}^{d_K}$, let $\tilde{x}^{(k)} = U_{k \to K}(D_{K \to k}(x))$. We also define the null image $\tilde{x}^{(0)} = \mathbf{0}$. Let the **detail vector** at level $k$ be $\tilde{\epsilon}^{(k)} = \tilde{x}^{(k)} - \tilde{x}^{(k-1)}$.

*Assumption* 1 (Orthogonality of Details). The detail vector $\tilde{\epsilon}^{(k)}$, which contains the frequency content for level $k$, is orthogonal to the detail vectors of all other levels.

$$\mathbb{E}[(\tilde{\epsilon}^{(k)})^T \tilde{\epsilon}^{(j)}] = 0 \quad \text{for} \quad k \neq j \tag{48}$$

This also reflects the nature of the process, in which each stage responsible for adding details is conditionally independent, as illustrated in equation 14. Under Assumption 1, the full image can be

written as a telescoping sum:

$$x = \tilde{x}^{(K)} = \sum_{k=1}^{K}(\tilde{x}^{(k)} - \tilde{x}^{(k-1)}) = \sum_{k=1}^{K}\tilde{\epsilon}^{(k)} \tag{49}$$

**Lemma 1** (Energy Decomposition). *Under Assumption 1, the expected energy of an image is the sum of the expected energies of its orthogonal detail components.*

$$\mathbb{E}[|x|^2] = \mathbb{E}\left[\left|\sum_{k=1}^{K}\tilde{\epsilon}^{(k)}\right|^2\right] = \sum_{k=1}^{K}\mathbb{E}[|\tilde{\epsilon}^{(k)}|^2] \tag{50}$$

*Proof.* This follows directly from the linearity of expectation and the orthogonality assumption, as all cross-terms $\mathbb{E}[(\tilde{\epsilon}^{(k)})^T \tilde{\epsilon}^{(j)}]$ for $k \neq j$ vanish. □

### G.1.3 ANALYSIS OF TRANSPORTATION COSTS

**Cost of the Direct Model** ($L_A$). The cost is $L_A = \mathbb{E}[|x_1 - x_0|^2]$, where $x_1 \sim x^{(K)}$ and $x_0 \sim \mathcal{N}(0, \sigma^2 I_{d_K})$. Due to independence and $\mathbb{E}[x_0] = 0$:

$$\begin{aligned} L_A &= \mathbb{E}[|x_1|^2] - 2\mathbb{E}[x_1^T x_0] + \mathbb{E}[|x_0|^2] \\ &= \mathbb{E}[|x_1|^2] + \mathbb{E}[|x_0|^2] \end{aligned} \tag{51}$$

Using Lemma 1 and the variance of the noise, $\mathbb{E}[|x_0|^2] = d_K\sigma^2$:

$$L_A = \left(\sum_{k=1}^{K}\mathbb{E}[|\tilde{\epsilon}^{(k)}|^2]\right) + d_K\sigma^2 \tag{52}$$

**Cost of the Cascaded Model with conditional dependent couplings** ($L_B$). The total cost is $L_B = \sum_{k=1}^{K} L_k$.

- **Stage 1:** $L_1 = \mathbb{E}[|x_1^{(1)} - x_0^{(1)}|^2] = \mathbb{E}[|x_1^{(1)}|^2] + \mathbb{E}[|x_0^{(1)}|^2]$. The data $x_1^{(1)}$ represents the coarsest level of detail, corresponding to $\tilde{\epsilon}^{(1)}$. The noise variance is $\mathbb{E}[|x_0^{(1)}|^2] = d_1\sigma^2$. Thus,

$$L_1 \approx \mathbb{E}[|\tilde{\epsilon}^{(1)}|^2] + d_1\sigma^2 \tag{53}$$

- **Stages** $k \in [2, K]$**:** The cost is $L_k = \mathbb{E}[|x_1^{(k)} - x_0^{(k)}|^2]$. Substituting the data-dependent coupling $x_0^{(k)} = U_k(D_k(x_1^{(k)})) + \sigma_k\zeta_k$, we get:

$$\begin{aligned} L_k &= \mathbb{E}[|x_1^{(k)} - U_k(D_k(x_1^{(k)})) - \sigma_k\zeta_k|^2] \\ &= \mathbb{E}[|x_1^{(k)} - U_k(D_k(x_1^{(k)}))|^2] - 2\,\mathbb{E}[\sigma_k\zeta_k^T \cdot (x_1^{(k)} - U_k(D_k(x_1^{(k)})))] + \mathbb{E}[|\sigma_k\zeta_k|^2] \\ &= \mathbb{E}[|x_1^{(k)} - U_k(D_k(x_1^{(k)}))|^2] - 2\,\mathbb{E}[\sigma_k\zeta_k]\,\mathbb{E}[(x_1^{(k)} - U_k(D_k(x_1^{(k)})))] + \mathbb{E}[|\sigma_k\zeta_k|^2] \\ &= \mathbb{E}[|x_1^{(k)} - U_k(D_k(x_1^{(k)}))|^2] + \mathbb{E}[|\sigma_k\zeta_k|^2], \end{aligned} \tag{54}$$

  where independence: $\sigma_k\zeta_k \perp\!\!\!\perp (x_1^{(k)} - U_k(D_k(x_1^{(k)})))$ and $\mathbb{E}[\sigma_k\zeta_k] = 0$. $\mathbb{E}[|x_1^{(k)} - U_k(D_k(x_1^{(k)}))|^2]$ is the expected energy of the details required to go from resolution $k-1$ to $k$, which corresponds precisely to the energy of the detail vector $\tilde{\epsilon}^{(k)}$: $\mathbb{E}[|\tilde{\epsilon}^{(k)}|^2]$, and the variance of the noise, $\mathbb{E}[|\sigma_k\zeta_k|^2] = d_k\sigma_k^2$:

$$L_k \approx \mathbb{E}[|\tilde{\epsilon}^{(k)}|^2] + d_k\sigma_k^2 \tag{55}$$

Summing the costs for all stages of Model B:

$$L_B = L_1 + \sum_{k=2}^{K} L_k$$

$$\approx (\mathbb{E}[|\tilde{\epsilon}^{(1)}|^2] + d_1\sigma^2) + \sum_{k=2}^{K} \left(\mathbb{E}[|\tilde{\epsilon}^{(k)}|^2] + d_k\sigma_k^2\right) \quad (56)$$

$$= \left(\sum_{k=1}^{K} \mathbb{E}[|\tilde{\epsilon}^{(k)}|^2]\right) + d_1\sigma^2 + \left(\sum_{k=2}^{K} d_k\sigma_k^2\right),$$

Here, $d_k = 2^{k-1}d_1$ for $1 \leq k \leq K$, reflecting the pyramid structure that aligns with the conditional dependent coupling design. As discussed in the previous section, we define $\sigma_k$ to decrease as $k$ increases, which matches the intuition that as the stage $k$ progresses, the requirements for diversity and stochasticity gradually diminish, as shown in equation 20, where we specify $\gamma = 2$ for simplicity:

$$\sigma_k = 2^{-(k-1)}\sigma \quad \text{for } 2 \leq k \leq K. \quad (57)$$

### G.1.4  MAIN PROOF AND CONCLUSION

*Proof.* From the detailed analysis in the previous subsections, we have derived the transportation costs for the direct and cascaded models:

$$L_A = \left(\sum_{k=1}^{K} \mathbb{E}[|\tilde{\epsilon}^{(k)}|^2]\right) + d_K\sigma^2, \quad (58)$$

where $d_K$ is the dimensionality of the high-resolution image space.

For the cascaded model with $K$ stages, the total cost is the sum of the costs of all stages:

$$L_B = \left(\sum_{k=1}^{K} \mathbb{E}[|\tilde{\epsilon}^{(k)}|^2]\right) + d_1\sigma^2 + \sum_{k=2}^{K} d_k\sigma_k^2. \quad (59)$$

The term $\sum_{k=1}^{K} \mathbb{E}[|\tilde{\epsilon}^{(k)}|^2]$, representing the total signal energy (see Lemma 1), appears in both $L_A$ and $L_B$ and thus cancels when we consider the difference:

$$L_A - L_B = d_K\sigma^2 - \left(d_1\sigma^2 + \sum_{k=2}^{K} d_k\sigma_k^2\right). \quad (60)$$

To evaluate the sign of $L_A - L_B$, we use the definitions of $d_k$ and $\sigma_k$ from the cascaded model with conditional dependent coupling:

$$d_k = 2^{k-1}d_1, \quad \sigma_k = 2^{-(k-1)}\sigma \quad \text{for } 2 \leq k \leq K.$$

Thus, the second term can be expanded as:

$$\sum_{k=2}^{K} d_k\sigma_k^2 = \sum_{k=2}^{K} \left(2^{k-1}d_1\right)\left(2^{-(k-1)}\sigma\right)^2 = d_1\sigma^2 \sum_{k=2}^{K} 2^{-(k-1)}. \quad (61)$$

We have:

$$L_A - L_B = \sigma^2 \left(d_K - d_1 - d_1 \sum_{k=2}^{K} 2^{-(k-1)}\right). \quad (62)$$

Since $d_K = 2^{K-1}d_1$, we obtain:

$$L_A - L_B = \sigma^2 d_1 \left(2^{K-1} - 1 - \sum_{k=2}^{K} 2^{-(k-1)}\right). \quad (63)$$

Specifically, $L_A - L_B = 0$ when $K = 1$. Observe that $2^{K-1} - 1 \gg \sum_{k=2}^{K} 2^{-(k-1)}$ for all $K \geq 2$, because the left-hand side grows exponentially while the right-hand side is a bounded geometric series:

$$\sum_{k=2}^{K} 2^{-(k-1)} < \sum_{m=1}^{\infty} 2^{-m} = 1.$$

Therefore, the term in parentheses is strictly positive:

$$2^{K-1} - 1 - \sum_{k=2}^{K} 2^{-(k-1)} > 0.$$

Since $\sigma^2 > 0$ and $d_1 > 0$, we conclude:

$$L_A - L_B > 0 \implies L_A > L_B. \tag{64}$$

This completes the proof. The main factor leading to $L_A > L_B$ is that the direct model pays the cost of transporting a high-dimensional Gaussian noise vector of dimension $d_K$, while the cascaded model introduces noise primarily in the low-dimensional early stages and relies on conditional dependent couplings to incrementally add fine-grained details. This multi-resolution structure results in a strictly lower total transportation cost. $\qquad \square$

$\square$

### G.2 Proof of Theorem 3

We provide a formal proof that the cascaded Flow Matching model with the **conditional dependent coupling** strategy has strictly smaller expected inference time than the naive single-stage model for high-resolution image generation, as stated in Theorem 3. The argument follows three steps: (i) relate the expected inference-time functional $\iota(\cdot)$ to the Number of Function Evaluations (NFE) and the per-evaluation cost, (ii) connect NFE to a transportation cost, and (iii) compare the resulting computational "loads" of the direct and cascaded schemes by separating signal and noise contributions.

#### G.2.1 Problem Formulation and Assumptions

Recall the theorem's notation:

$$T_A = \mathbb{E}\big[\iota\big(x_0 \to \hat{x}_1\big)\big], \qquad T_B = \sum_{k=1}^{K} \mathbb{E}\Big[\iota\big(x_0^{(k)} \to \hat{x}_1^{(k)}\big)\Big].$$

The direct model draws $x_0 \sim \mathcal{N}(0, \sigma^2 I_{d_K})$ and produces $\hat{x}_1 \in \mathbb{R}^{d_K}$ in one stage; the cascaded model comprises $K$ stages, with stage $k$ transporting $x_0^{(k)} \sim \mathcal{N}(x_0^{(k)}; m_k(x_1^{(k)}), \sigma_k^2 I_{d_k})$ to $\hat{x}_1^{(k)} \in \mathbb{R}^{d_k}$, where $d_1 < \cdots < d_K$, as state in equation 10.

**Per-evaluation cost model.** For ODE-based generators (e.g., probability flow ODEs), the expected inference-time cost is the product of the NFE and the cost per function evaluation. With a fixed backbone and solver tolerance, the per-evaluation cost scales linearly with the ambient dimensionality:

$$\mathbb{E}\big[\iota\big(x_0 \to \hat{x}_1\big)\big] = \alpha\, N_A\, d_K, \qquad \mathbb{E}\Big[\iota\big(x_0^{(k)} \to \hat{x}_1^{(k)}\big)\Big] = \alpha\, N_k\, d_k,$$

for some constant $\alpha > 0$. Hence

$$T_A = \alpha\, N_A\, d_K, \qquad T_B = \alpha \sum_{k=1}^{K} N_k\, d_k.$$

To prove $T_A > T_B$ it suffices to show

$$\sum_{k=1}^{K} N_k\, d_k < N_A\, d_K. \tag{65}$$

*Assumption* 2 (NFE vs. Transportation Cost). The NFE required to solve a probability flow ODE to a given accuracy is proportional to the transportation cost $L = \mathbb{E}[|x_1 - x_0|^2]$. This cost serves as a proxy for the complexity of the vector field.

$$N_i = C \cdot L_i \tag{66}$$

where $C$ is a constant dependent on the ODE solver and desired precision.

Under this assumption, equation 65 is equivalent to

$$\sum_{k=1}^{K} L_k \, d_k \; < \; L_A \, d_K. \tag{67}$$

Define the *computational load* of a transport as $\text{Load} := L \times d$. Then

$$\text{Load}_A \;=\; L_A d_K, \qquad \text{Load}_B \;=\; \sum_{k=1}^{K} L_k d_k,$$

and proving equation 67 is equivalent to showing $\text{Load}_B < \text{Load}_A$.

### G.2.2 ANALYSIS OF COMPUTATIONAL LOAD

We use the transportation-cost decompositions established in Section G.1. Writing $\tilde{\epsilon}^{(k)}$ for the signal component associated with scale/stage $k$:

**Direct model (single stage).** The load is the product of its transportation cost $L_A$(equation 58) and the dimensionality of the final image $d_K$.

$$
\begin{aligned}
\text{Load}_A = L_A \cdot d_K \\
= \left( \left( \sum_{j=1}^{K} \mathbb{E}[|\tilde{\epsilon}^{(j)}|^2] \right) + d_K \sigma^2 \right) d_K \\
= d_K \sum_{j=1}^{K} \mathbb{E}[|\tilde{\epsilon}^{(j)}|^2] + d_K^2 \sigma^2
\end{aligned}
\tag{68}
$$

**Cascaded model (multi-stage).** The total load is the sum of the loads of each stage, where the load for stage $k$ is $L_k \cdot d_k$, where $L_k$ is shown in equation 55. For stage 1, $x_0^{(1)} \sim \mathcal{N}(0, \sigma^2 I_{d_1})$; for stages $k \geq 2$, the conditional dependent coupling strategy injects $x_0^{(k)} \sim \mathcal{N}(x_0^{(k)}; m_k(x_1^{(k)}), \sigma_k^2 I_{d_k})$. Hence

$$
\begin{aligned}
\text{Load}_B &= \sum_{k=1}^{K} L_k d_k = L_1 d_1 + \sum_{k=2}^{K} L_k d_k \\
&= (\mathbb{E}[|\tilde{\epsilon}^{(1)}|^2] + d_1 \sigma^2) d_1 + \sum_{k=2}^{K} (\mathbb{E}[|\tilde{\epsilon}^{(k)}|^2] + d_k \sigma_k^2) d_k \\
&= \mathbb{E}[|\tilde{\epsilon}^{(1)}|^2] d_1 + d_1^2 \sigma^2 + \sum_{k=2}^{K} \mathbb{E}[|\tilde{\epsilon}^{(k)}|^2] d_k + \sum_{k=2}^{K} d_k^2 \sigma_k^2 \\
&= \left( \sum_{k=1}^{K} \mathbb{E}[|\tilde{\epsilon}^{(k)}|^2] d_k \right) + \left( d_1^2 \sigma^2 + \sum_{k=2}^{K} d_k^2 \sigma_k^2 \right)
\end{aligned}
\tag{69}
$$

### G.2.3 MAIN PROOF AND CONCLUSION

We compare equation 68 and equation 69 by splitting into signal and noise parts.

**(1) Signal component.** From equation 68 and equation 69,

$$\text{Load}_{A,\text{signal}} = d_K \sum_{k=1}^{K} \mathbb{E}\big[|\tilde{\epsilon}^{(k)}|^2\big], \qquad \text{Load}_{B,\text{signal}} = \sum_{k=1}^{K} \mathbb{E}\big[|\tilde{\epsilon}^{(k)}|^2\big]\, d_k.$$

Since $d_k < d_K$ for all $1 \le k < K$, each term in the summation for $\text{Load}_{B,\text{signal}}$ is smaller than(or equal to, when $k = K$) the corresponding term for $\text{Load}_{A,\text{signal}}$. Therefore:

$$\sum_{k=1}^{K} \mathbb{E}[|\tilde{\epsilon}^{(k)}|^2]d_k < d_K \sum_{k=1}^{K} \mathbb{E}[|\tilde{\epsilon}^{(k)}|^2] \tag{70}$$

The computational work related to generating the image content is strictly lower in the cascaded model because most detail components $\tilde{\epsilon}^{(k)}$ are processed at much lower dimensionalities $d_k$.

**(2) Noise component (coupling strategy).** Adopt the conditional dependent coupling in which spatial resolutions double by stage and the injected noise variance is inversely scaled, as shown in equation 20:

$$d_k = 2^{k-1}d_1, \quad \sigma_k = 2^{-(k-1)}\sigma \quad \text{for } 2 \le k \le K.$$

Then

$$
\begin{aligned}
\text{Load}_{B,\text{noise}} &= d_1^2\sigma^2 + \sum_{k=2}^{K} d_k^2\sigma_k^2 \\
&= d_1^2\sigma^2 + \sum_{k=2}^{K} (2^{k-1}d_1)^2(2^{-(k-1)}\sigma)^2 \\
&= d_1^2\sigma^2 + \sum_{k=2}^{K} (2^{2(k-1)}d_1^2)(2^{-2(k-1)}\sigma^2) \\
&= d_1^2\sigma^2 + \sum_{k=2}^{K} d_1^2\sigma^2 \\
&= d_1^2\sigma^2 + (K-1)d_1^2\sigma^2 = K d_1^2\sigma^2,
\end{aligned}
\tag{71}
$$

whereas for the direct model

$$\text{Load}_{A,\text{noise}} = d_K^2\sigma^2 = (2^{K-1}d_1)^2\sigma^2 = 2^{2(K-1)}d_1^2\sigma^2 \tag{72}$$

For $K = 1$, both loads coincide; for any practical cascade with $K \ge 2$,

$$\text{Load}_{B,\text{noise}} = K d_1^2\sigma^2 < 2^{2(K-1)}d_1^2\sigma^2 = \text{Load}_{A,\text{noise}}. \tag{73}$$

Thus the noise-related work is exponentially smaller in the cascaded scheme.

**Combining (1) and (2).** Adding equation 70 and equation 73 yields

$$\text{Load}_B < \text{Load}_A \iff \sum_{k=1}^{K} L_k d_k < L_A d_K.$$

Using the NFE–transportation-cost proportionality and the per-evaluation cost model, common multiplicative constants cancel, giving

$$T_B = \sum_{k=1}^{K} \mathbb{E}\Big[\iota\big(x_0^{(k)} \to \hat{x}_1^{(k)}\big)\Big] < \mathbb{E}\big[\iota\big(x_0 \to \hat{x}_1\big)\big] = T_A.$$

For $K = 1$ the two procedures coincide, while for every $K \ge 2$ the inequality is strict. This completes the proof of Theorem 3.

# H  ALGORITHM

Here we present the detailed unconditional training and inference procedures for the unified multi-stage generative model $b^\theta$ with **conditional dependent coupling** in Algorithm 1 and Algorithm 2.

---

**Algorithm 1** Training a unified multi-stage Flow-Matching model $b^\theta$ (ODE view)

---

**Require:** Dataset $\mathcal{D} = \{x^{(K)}\} \sim p_{\text{data}}$, stages $k \in \{1, \ldots, K\}$ with dimensions $d_k$, feature-map resolutions $r_k$ and resolution embedding $e_k = E(r_k)$; down/upsamplers $D_k : \mathbb{R}^{d_k} \to \mathbb{R}^{d_{k-1}}$ and $U_k : \mathbb{R}^{d_{k-1}} \to \mathbb{R}^{d_k}$, composite downsampler $D_{K \to k}$; time partition $0 = t_0^1 < t_1^1 = t_0^2 < \cdots < t_1^K = 1$; noise scales $\sigma$ (stage 1) and $\{\sigma_k\}_{k=2}^K$; batch size $B$.

    **Comment:** Ground-truth at stage $k$ is $x_1^{(k)} = D_{K \to k}(x^{(K)}) \sim \rho_1^{(k)}(\cdot \mid x^{(K)})$.

1: **for** each training step **do**
2:     **for** $i = 1$ **to** $B$ **do**
3:         Sample $x^{(K)} \sim p_{\text{data}}$; sample $t \sim \mathcal{U}[0, 1]$.
4:         Find $k$ such that $t \in [t_0^k, t_1^k]$; set the rescaled time $\tau \leftarrow \dfrac{t - t_0^k}{t_1^k - t_0^k} \in [0, 1]$.
5:         **Target at stage $k$:** $x_1^{(k)} \leftarrow D_{K \to k}(x^{(K)})$         $\triangleright\ x_1^{(k)} \sim \rho_1^{(k)}(\cdot \mid x^{(K)})$
6:         **if** $k = 1$ **then**         $\triangleright$ Stage 1: noise-to-image coupling is independent
7:             Sample $x_0^{(1)} \sim \mathcal{N}(0, \sigma^2 I_{d_1})$         $\triangleright\ \rho_0^{(1)}(x_0^{(1)})$
8:         **else**         $\triangleright$ Stage $k > 1$: conditional dependent coupling
9:             Sample $\zeta_k \sim \mathcal{N}(0, I_{d_k})$
10:           $m_k(x_1^{(k)}) \leftarrow U_k\big(D_k(x_1^{(k)})\big)$
11:           $x_0^{(k)} \leftarrow m_k(x_1^{(k)}) + \sigma_k \zeta_k$         $\triangleright\ \rho_0^{(k)}(\cdot \mid x_1^{(k)}) = \mathcal{N}(m_k, \sigma_k^2 I)$
12:         **end if**
13:         **Linear interpolant:** $I_\tau^{(k)} \leftarrow (1 - \tau) x_0^{(k)} + \tau x_1^{(k)}$
14:         **Target velocity:** $\dot{I}_\tau^{(k)} \leftarrow x_1^{(k)} - x_0^{(k)}$         $\triangleright$ constant in $\tau$ for linear path
15:         $e_k \leftarrow E(r_k)$         $\triangleright$ resolution embedding
16:         $u_\theta \leftarrow b^\theta(I_\tau^{(k)}, \tau, e_k)$         $\triangleright$ DiT vector field shared across stages
17:         Per-sample loss: $\ell_i \leftarrow \|u_\theta\|_2^2 - 2\,\dot{I}_\tau^{(k)} \cdot u_\theta$         $\triangleright$ equiv. to $\|u_\theta - \dot{I}_\tau^{(k)}\|_2^2$ up to a const.
18:     **end for**
19:     $\mathcal{L}(\theta) \leftarrow \frac{1}{B} \sum_{i=1}^B \ell_i$         $\triangleright$ matches $\mathbb{E}\big[\|b^\theta\|^2 - 2\,\dot{I} \cdot b^\theta\big]$
20:     Update parameters: $\theta \leftarrow \theta - \eta \nabla_\theta \mathcal{L}(\theta)$
21: **end for**

---

The inference steps are described using the forward Euler method Lipman et al. (2024; 2022) for solving the ODE, as a simple example of a numerical solver.

## I    CLASSIFIER-FREE GUIDANCE FOR CONDITIONAL DEPENDENT COUPLING

**Rationale.** Because each stage $k$ in our multi-stage model learns the *accurate* data distribution $\rho_1^{(k)}(x_1^{(k)} \mid x^{(K)})$ under the **conditional dependent coupling** (Sec. 3.3), Classifier-Free Guidance (CFG) Ho & Salimans (2022) can be applied *uniformly* across stages. This obviates complex stage-wise guidance schedules Chen et al. (2025); Kynkäänniemi et al. (2024) and yields a simpler, robust implementation.

**Training with conditional dropout (classifier-free).** We train the unified DiT $b^\theta$ exactly as in the conditional generation objective shown in equation 19, with the only change that the conditioning signal is randomly dropped. Let $\mathbf{c}$ denote the conditioning signal (e.g., class label or text prompt), and let $\varnothing$ denote the null condition. Define the mixed conditioning

$$\bar{\mathbf{c}} \sim q(\bar{\mathbf{c}}) = (1 - p_\varnothing)\,p(\mathbf{c}) + p_\varnothing\,\delta_\varnothing, \tag{74}$$

where $p_\varnothing \in (0, 1)$ is the dropout probability and $\delta_\varnothing$ is a point mass at $\varnothing$. The training loss becomes

$$\mathcal{L}_{\text{cfg}}(\theta) = \mathbb{E}_{\substack{t \sim [0,1],\, k,\, e_k, \\ (x_0^{(k)}, x_1^{(k)}) \sim \rho^{(k)}(x_0^{(k)}, x_1^{(k)} | x^{(K)}), \\ \bar{\mathbf{c}} \sim q}} \Big[ \big\|b^\theta(I_\tau^{(k)}, \tau, e_k, \bar{\mathbf{c}})\big\|^2 - 2\,\dot{I}_\tau^{(k)} \cdot b^\theta(I_\tau^{(k)}, \tau, e_k, \bar{\mathbf{c}}) \Big],$$

$$\tag{75}$$

with $I_\tau^{(k)} = (1 - \tau)x_0^{(k)} + \tau x_1^{(k)}$, $\dot{I}_\tau^{(k)} = x_1^{(k)} - x_0^{(k)}$, and $\tau = \frac{t - t_0^k}{t_1^k - t_0^k}$ as in Sec. 3.3.

---

**Algorithm 2** Inference via sequential multi-stage ODE integration (forward Euler)

---

**Require:** Trained vector field $b^\theta$; stages $k \in \{1, \ldots, K\}$ with dimensions $d_k$, feature-map resolutions $r_k$ and resolution embedding $e_k = E(r_k)$; upsamplers $U_k : \mathbb{R}^{d_{k-1}} \to \mathbb{R}^{d_k}$; noise scales $\sigma$ (stage 1) and $\{\sigma_k\}_{k=2}^K$; per-stage step counts $\{N_k\}$.

    **Comment:** The Markovian cascade uses $\hat{x}_0^{(1)} \sim \mathcal{N}(0, \sigma^2 I_{d_1})$ and, for $k \geq 2$, $\hat{x}_0^{(k)} = U_k(\hat{x}_1^{(k-1)}) + \sigma_k \zeta_k$ with $\zeta_k \sim \mathcal{N}(0, I_{d_k})$.

  1: **Initialize stage 1 (noise-to-image):** Sample $\hat{x}_0^{(1)} \sim \mathcal{N}(0, \sigma^2 I_{d_1})$; set $\hat{X}_0^{(1)} \leftarrow \hat{x}_0^{(1)}$.
  2: **for** $k = 1$ **to** $K$ **do**
  3:     $\Delta\tau \leftarrow 1/N_k$; $e_k \leftarrow E(r_k)$
  4:     **if** $k > 1$ **then**                            ▷ Conditional dependent coupling from previous stage
  5:         Sample $\zeta_k \sim \mathcal{N}(0, I_{d_k})$
  6:         $\hat{x}_0^{(k)} \leftarrow U_k(\hat{x}_1^{(k-1)}) + \sigma_k \zeta_k$
  7:         $\hat{X}_0^{(k)} \leftarrow \hat{x}_0^{(k)}$
  8:     **end if**
  9:     **for** $n = 0$ **to** $N_k - 1$ **do**
 10:         $\tau \leftarrow n\,\Delta\tau$
 11:         $\hat{v} \leftarrow b^\theta(\hat{X}_\tau^{(k)}, \tau, e_k)$                   ▷ ODE velocity at rescaled time $\tau \in [0, 1]$
 12:         **Euler step:** $\hat{X}_{\tau+\Delta\tau}^{(k)} \leftarrow \hat{X}_\tau^{(k)} + \Delta\tau \cdot \hat{v}$
 13:     **end for**
 14:     **Stage output:** $\hat{x}_1^{(k)} \leftarrow \hat{X}_1^{(k)}$                          ▷ resolution $d_k$
 15: **end for**
 16: **return** $\hat{x}_1^{(K)} \in \mathbb{R}^{d_K}$                           ▷ final high-resolution sample

---

**CFG at inference: guided vector field.** Let the *conditional* and *unconditional* vector-field predictions be

$$b_{\mathbf{c}}^\theta(z, \tau, e_k) = b^\theta(z, \tau, e_k, \mathbf{c}), \qquad b_\varnothing^\theta(z, \tau, e_k) = b^\theta(z, \tau, e_k, \varnothing). \tag{76}$$

Given a global guidance scale $S_{\mathrm{cfg}} \geq 0$ that is *shared across all stages* $k$, we define the CFG-guided field by the standard linear rule Ho & Salimans (2022):

$$\begin{aligned} b_{\mathrm{cfg}}^\theta(z, \tau, e_k; \mathbf{c}, S_{\mathrm{cfg}}) &= b_\varnothing^\theta(z, \tau, e_k) + S_{\mathrm{cfg}}\Big(b_{\mathbf{c}}^\theta(z, \tau, e_k) - b_\varnothing^\theta(z, \tau, e_k)\Big) \\ &= (1 - S_{\mathrm{cfg}})\, b_\varnothing^\theta + S_{\mathrm{cfg}}\, b_{\mathbf{c}}^\theta. \end{aligned} \tag{77}$$

This recovers unconditional generation when $S_{\mathrm{cfg}} = 0$, the nominal conditional model when $S_{\mathrm{cfg}} = 1$, and stronger condition-following for $S_{\mathrm{cfg}} > 1$.

**Stage-wise conditional generation under coupling.** At inference, the state $X_\tau^{(k)}$ in stage $k$ evolves under the guided ODE driven by equation 77:

$$\frac{\mathrm{d}}{\mathrm{d}\tau} X_\tau^{(k)} = b_{\mathrm{cfg}}^\theta(X_\tau^{(k)}, \tau, e_k; \mathbf{c}, S_{\mathrm{cfg}}), \qquad \tau \in [0, 1], \tag{78}$$

with initial condition

$$X_0^{(1)} = x_0^{(1)} \sim \mathcal{N}(0, \sigma^2 I_{d_1}), \qquad X_0^{(k)} = U_k(\hat{x}_1^{(k-1)}) + \sigma_k \zeta_k, \;\; \zeta_k \sim \mathcal{N}(0, I_{d_k}), \;\; k \geq 2, \tag{79}$$

where the conditional dependent coupling is exactly the construction in equation 9. The stage output is the terminal state

$$\hat{x}_1^{(k)} = X_1^{(k)} = X_0^{(k)} + \int_0^1 b_{\mathrm{cfg}}^\theta(X_t^{(k)}, t, e_k; \mathbf{c}, S_{\mathrm{cfg}})\, \mathrm{d}t, \tag{80}$$

and the full cascade is obtained by the recursion

$$\hat{x}_1^{(1)} = X_1^{(1)}, \qquad \hat{x}_1^{(k)} = X_1^{(k)}\big(U_k(\hat{x}_1^{(k-1)}) + \sigma_k \zeta_k\big), \;\; k = 2, \ldots, K, \tag{81}$$

which is the same Markov structure as equation 16 but with the guided field $b_{\mathrm{cfg}}^\theta$.

**Remarks.** (i) The use of a *single* $S_{\mathrm{cfg}}$ across all stages is justified by the fact that, under conditional dependent coupling, each stage already targets the correct $\rho_1^{(k)}(x_1^{(k)} \mid x^{(K)})$. Hence, CFG

serves primarily to bias the direction of transport within each accurately learned stage rather than to compensate for a stage mismatch, removing the need for stage-wise tuning. (ii) For an SDE parameterization, the same rule equation 77 is applied to the predicted score/drift in place of $b^\theta$:

$$s^\theta_{\text{cfg}}(z, t, e_k; \mathbf{c}, S_{\text{cfg}}) = s^\theta_\varnothing(z, t, e_k) + S_{\text{cfg}}\big(s^\theta_{\mathbf{c}}(z, t, e_k) - s^\theta_\varnothing(z, t, e_k)\big),$$

and the sampler integrates the corresponding guided stochastic dynamics with $X_0^{(k)}$ as in equation 79. (iii) Following the CFG strategy introduced in this section for conditional generation, we extend the unconditional generation procedure described in Appendix §H, and provide the training and inference procedures of the unified multi-stage generative model $b^\theta$ with CFG as shown below.

---

**Algorithm 3** Training a unified multi-stage Flow-Matching model $b^\theta$ with Classifier-Free Guidance

---

**Require:** Dataset $\mathcal{D} = \{x^{(K)}, \mathbf{c}\}$ with conditions $\mathbf{c}$ (e.g., class/text); stages $k \in \{1, \dots, K\}$ with dimensions $d_k$, feature-map resolutions $r_k$ and resolution embedding $e_k = E(r_k)$; down/upsamplers $D_k : \mathbb{R}^{d_k} \to \mathbb{R}^{d_{k-1}}$ and $U_k : \mathbb{R}^{d_{k-1}} \to \mathbb{R}^{d_k}$; composite downsampler $D_{K \to k}$; time partition $0 = t_0^1 < t_1^1 = t_0^2 < \cdots < t_1^K = 1$; noise scales $\sigma$ (stage 1) and $\{\sigma_k\}_{k=2}^K$; classifier-free dropout probability $p_\varnothing \in (0, 1)$; batch size $B$.

    **Comment:** Ground-truth at stage $k$ is $x_1^{(k)} = D_{K \to k}(x^{(K)}) \sim \rho_1^{(k)}(\cdot \mid x^{(K)})$. Conditional dropout samples $\bar{\mathbf{c}}$ from $q(\bar{\mathbf{c}}) = (1 - p_\varnothing)\, p(\mathbf{c}) + p_\varnothing\, \delta_\varnothing$.

1: **for** each training step **do**
2:     **for** $i = 1$ **to** $B$ **do**
3:         Sample $(x^{(K)}, \mathbf{c}) \sim \mathcal{D}$; sample $t \sim \mathcal{U}[0, 1]$.
4:         Find $k$ such that $t \in [t_0^k, t_1^k]$; set $\tau \leftarrow \dfrac{t - t_0^k}{t_1^k - t_0^k} \in [0, 1]$.
5:         **Target at stage $k$:** $x_1^{(k)} \leftarrow D_{K \to k}(x^{(K)})$.
6:         **if** $k = 1$ **then**         ▷ Stage 1: noise-to-image, independent prior
7:             Sample $x_0^{(1)} \sim \mathcal{N}(0, \sigma^2 I_{d_1})$         ▷ $\rho_0^{(1)}(x_0^{(1)})$
8:         **else**         ▷ Stage $k > 1$: conditional dependent coupling
9:             Sample $\zeta_k \sim \mathcal{N}(0, I_{d_k})$
10:           $m_k(x_1^{(k)}) \leftarrow U_k\big(D_k(x_1^{(k)})\big)$
11:           $x_0^{(k)} \leftarrow m_k(x_1^{(k)}) + \sigma_k \zeta_k$         ▷ $\rho_0^{(k)}(\cdot \mid x_1^{(k)}) = \mathcal{N}(m_k, \sigma_k^2 I)$
12:         **end if**
13:         **Linear interpolant:** $I_\tau^{(k)} \leftarrow (1 - \tau)\, x_0^{(k)} + \tau\, x_1^{(k)}$
14:         **Target velocity:** $\dot{I}_\tau^{(k)} \leftarrow x_1^{(k)} - x_0^{(k)}$         ▷ constant in $\tau$ for linear path
15:         **Conditional dropout:** Draw $u \sim \mathcal{U}[0, 1]$; set $\bar{\mathbf{c}} \leftarrow \varnothing$ if $u < p_\varnothing$, else $\bar{\mathbf{c}} \leftarrow \mathbf{c}$.
16:         $e_k \leftarrow E(r_k)$         ▷ resolution embedding fused with time embedding
17:         $u_\theta \leftarrow b^\theta\big(I_\tau^{(k)}, \tau, e_k, \bar{\mathbf{c}}\big)$         ▷ DiT vector field shared across stages
18:         Per-sample loss: $\ell_i \leftarrow \|u_\theta\|_2^2 - 2\, \dot{I}_\tau^{(k)} \cdot u_\theta$         ▷ matches equation 75
19:     **end for**
20:     $\mathcal{L}_{\text{cfg}}(\theta) \leftarrow \frac{1}{B} \sum_{i=1}^B \ell_i$
21:     Update parameters: $\theta \leftarrow \theta - \eta \nabla_\theta \mathcal{L}_{\text{cfg}}(\theta)$
22: **end for**

---

---

**Algorithm 4** Inference via sequential multi-stage ODE integration with Classifier-Free Guidance

---

**Require:** Trained vector field $b^\theta$; user condition $\mathbf{c}$; global guidance scale $S_{\text{cfg}} \geq 0$ (shared across stages); stages $k \in \{1, \ldots, K\}$ with dimensions $d_k$, feature-map resolutions $r_k$ and resolution embedding $e_k = E(r_k)$; upsamplers $U_k : \mathbb{R}^{d_{k-1}} \to \mathbb{R}^{d_k}$; noise scales $\sigma$ (stage 1) and $\{\sigma_k\}_{k=2}^K$; per-stage step counts $\{N_k\}$.
    **Comment:** Guided field $b_{\text{cfg}}^\theta(z, \tau, e_k; \mathbf{c}, S_{\text{cfg}}) = (1 - S_{\text{cfg}})b_\varnothing^\theta(z, \tau, e_k) + S_{\text{cfg}}b_{\mathbf{c}}^\theta(z, \tau, e_k)$, cf. equation 77.

1: **Initialize stage 1 (noise-to-image):** Sample $\hat{x}_0^{(1)} \sim \mathcal{N}(0, \sigma^2 I_{d_1})$; set $\hat{X}_0^{(1)} \leftarrow \hat{x}_0^{(1)}$.
2: **for** $k = 1$ **to** $K$ **do**
3:     $\Delta\tau \leftarrow 1/N_k$; $e_k \leftarrow E(r_k)$
4:     **if** $k > 1$ **then**                 $\triangleright$ Conditional dependent coupling from previous stage
5:         Sample $\zeta_k \sim \mathcal{N}(0, I_{d_k})$
6:         $\hat{x}_0^{(k)} \leftarrow U_k\big(\hat{x}_1^{(k-1)}\big) + \sigma_k \zeta_k$
7:         $\hat{X}_0^{(k)} \leftarrow \hat{x}_0^{(k)}$
8:     **end if**
9:     **for** $n = 0$ **to** $N_k - 1$ **do**
10:        $\tau \leftarrow n\,\Delta\tau$
11:        **Unconditional:** $\hat{v}_\varnothing \leftarrow b^\theta\big(\hat{X}_\tau^{(k)}, \tau, e_k, \varnothing\big)$
12:        **Conditional:** $\hat{v}_{\mathbf{c}} \leftarrow b^\theta\big(\hat{X}_\tau^{(k)}, \tau, e_k, \mathbf{c}\big)$
13:        **Guided field:** $\hat{v}_{\text{cfg}} \leftarrow (1 - S_{\text{cfg}})\hat{v}_\varnothing + S_{\text{cfg}}\hat{v}_{\mathbf{c}}$    $\triangleright$ $S_{\text{cfg}}{=}0$ uncond., $S_{\text{cfg}}{=}1$ nominal cond.
14:        **Euler step:** $\hat{X}_{\tau+\Delta\tau}^{(k)} \leftarrow \hat{X}_\tau^{(k)} + \Delta\tau \cdot \hat{v}_{\text{cfg}}$
15:     **end for**
16:     **Stage output:** $\hat{x}_1^{(k)} \leftarrow \hat{X}_1^{(k)}$                    $\triangleright$ resolution $d_k$
17: **end for**
18: **return** $\hat{x}_1^{(K)} \in \mathbb{R}^{d_K}$                    $\triangleright$ final high-resolution sample

## J   Implement Details

We report the implementation details of our model in Table 7.

All experiments are conducted on a platform equipped with $32 \times$ A100 GPUs (80GB). Under the configuration shown below, the global training batch size is $32 \times 8 = 256$. We train for 200 epochs; for ImageNet-1k, which contains 1.28M samples, each epoch consists of approximately 5000 steps. Thus, the full training schedule corresponds to roughly 1M iterations (a metric used in several prior works). With this setup, the total training time is approximately 76 hours, averaging 0.38 hours per epoch. During training, each A100 GPU consumes about 43GB of memory.

For inference, we use a single A100 GPU and set the batch size to 32 for all methods. We generate 50k images for evaluation and report inference speed in seconds per image.

## K   USE OF LARGE LANGUAGE MODELS (LLMS)

We acknowledge the use of a Generative AI tool for assistance with language editing and refinement during the preparation of this manuscript. The tool was not used for any substantive aspect of the research, such as ideation, data interpretation, or the formulation of conclusions. Full responsibility for all content rests with the authors.

Table 7: Complete Parameter Configuration for CDC-FM

| Category | Parameter | Value | Description |
|---|---|---|---|
| **Model Architecture** | num_attention_heads | 16 | Number of attention heads |
| | attention_head_dim | 72 | Dimension of each attention head |
| | in_channels | 3 | Input channels (RGB) |
| | out_channels | 3 | Output channels (RGB) |
| | depth | 28 | Number of transformer layers |
| | num_classes | 1000 | Number of ImageNet classes |
| | patch_size | 4 | Patch size for patch embedding |
| | attention_bias | true | Whether to use bias in attention |
| | embed_dim | 1152 | Embedding dimension ($16 \times 72$) |
| | dropout | 0.0 | Dropout rate |
| **Scheduler** | num_train_timesteps | 1000 | Number of training timesteps |
| | num_stages | 4 | Number of cascade stages |
| | diminish factor $\gamma$ | 2.0 | $\gamma$ in CDC-FM |
| **Training** | lr | 1e-4 | Learning rate |
| | weight_decay | 0.0 | Weight decay |
| | epochs | 200 | Number of training epochs |
| | grad_clip_norm | 1.0 | Gradient clipping norm |
| | ema_decay | 0.9999 | EMA decay rate |
| | logging_steps | 100 | Logging frequency |
| **Data** | root | /public/datasets/ILSVRC2012/train | Dataset root path |
| | center_crop | false | Whether to center crop |
| | resolution | 256 | Image resolution |
| | expand_ratio | 1.125 | Image expansion ratio |
| | num_workers | 8 | Number of data loader workers |
| | batch_size | 8 | Batch size per GPU |
| **Data Augmentation** | RandomHorizontalFlip | true | Random horizontal flip |
| | RandomCrop | true | Random crop (if not center crop) |
| | Resize | LANCZOS | Resize interpolation method |
| | Normalize | [0.5, 0.5, 0.5] | Normalization mean and std |
| **Optimization** | optimizer | AdamW | Optimizer type |
| | Optimizer Hyperparameters | $\beta_1 = 0.9,\ \beta_2 = 0.95,\ \epsilon = 1e^{-6}$ | Optimizer Hyperparameters |
| | Learning rate | $1e^{-4}$ | Learning rate |
| | precision | bfloat16 | Training precision |
| | deterministic_ops | false | Deterministic operations |
| **Inference** | num_inference_steps | 24 | Steps per stage (default) |
| | CFG strength $S_{cfg}$ | 3.0 | CFG scale (evaluation) |
| | num_fid_samples | 50000 | Number of FID samples |

