# OpenReview forum: "Stochastic Interpolants via Conditional Dependent Coupling"
_ICLR.cc/2026/Conference — Submitted to ICLR 2026_

### Official Review · Reviewer_Kt39 · 2025-10-27

**Soundness:** 2
**Presentation:** 2
**Contribution:** 3
**Rating:** 4
**Confidence:** 3

**Summary:**

This paper introduces Flow Matching with Conditional Dependent Coupling, a framework for image generation that operates directly in the pixel space. Instead of relying on a latent representation, the method models low-resolution images as being drawn from a Gaussian noise source and learns the transport process from low-resolution to high-resolution images. All stages are unified within a single DiT backbone and a resolution embedding as the condition for different stages. Compared to conventional pixel-space generation, Conditional Dependent Coupling significantly reduces transport cost and inference time. Experiments on ImageNet-1K at 256×256 and 512×512 resolutions demonstrate competitive FID and IS scores compared with state-of-the-art approaches.

**Strengths:**

1. The authors unify low-resolution generation and high-resolution upsampling within a single DiT backbone. Compared to previous multi-stage approaches, this design simplifies the overall framework and reduces engineering overhead.
2. The authors provide strong theoretical support for the effectiveness of conditional dependent coupling. In particular, they formally demonstrate that the proposed method reduces transport cost and inference computation cost, as established in Theorem 2 and Theorem 3.
3. The experimental results are competitive with other state-of-the-art methods operating directly in pixel space.

**Weaknesses:**

1. The paper does not provide sufficient information regarding the training cost, including hardware specifications, total training epochs, and computational resources. Although Table 3 in the Appendix includes some configuration details, it appears mislabeled as PixelFlow, suggesting potential inconsistencies. Moreover, the reported training duration is only 10 epochs, which is unusually short for this line of work, where diffusion-based methods typically require 400–1600 epochs on ImageNet. If accurate, this represents an approximately 40× to 160x reduction in training cost, which would be a remarkable advantage and thus requires clearer justification and verification.

2. The paper does not include ablation studies that quantify the contribution of key design components. While the model comprises K flow-matching stages, the authors only report inference cost per stage, without analyzing how each stage affects the final image quality. Additionally, there are no visualizations of intermediate outputs across stages, which would provide valuable insights into the effectiveness of the multi-stage transport process. A direct comparison between single-stage (direct pixel modeling) and multi-stage settings is also missing—this experiment would serve as the most direct empirical evidence supporting the core claim that the proposed coupling design, rather than the training recipe or backbone choice, drives the improvement.

3. Although inference speed is presented as one of the main contributions, the paper lacks a clear description of how “Speed” in Table 1 is defined or measured. Furthermore, comparisons with stronger recent baselines such as REPA (FID = 1.42) [1] and LightingDiT (FID = 1.35) [2] are missing, even though these methods achieve superior performance with comparable model sizes. Including such baselines would provide a more convincing demonstration of the proposed method’s practical advantage.

[1] Representation Alignment for Generation: Training Diffusion Transformers Is Easier Than You Think, ICLR 2025.
[2] Reconstruction vs. Generation: Taming Optimization Dilemma in Latent Diffusion Models, CVPR 2025.

**Questions:**

1. Could you clarify the actual training cost of the proposed methods including hardware specifications, total wall-clock time and number of training epochs for reproducibility?
2. Could you provide more ablation studies including the proposed improvement over previous one-stage pixel-wise methods or latent space-based methods?
3. How exactly is the 'Speed' metric in Table 1 defined and measured (Is it seconds per image)? Is the Speed in Table 1 evaluated under the same configuration (batch size, GPU type)?

---

> ### Author Response · Authors · 2025-11-23
> **Reply to Reviewer Kt39**
>
> # reply to Reviewer Kt39
>
> Thank you for the constructive feedback and thoughtful questions.
> Our responses to these points are provided below.
> **All corresponding revisions have been fully incorporated into the updated paper, colored in blue.**
>
> ---
>
> ### **W1, Q1. Reproducibility**
>
> The previous configuration table was exported directly from our debugging code bench.
> We have now updated it to include the full experimental configuration, as shown in Appendix J.
>
> ---
>
> ### **W2. Visualizations of intermediate outputs across stages**
>
> See Appendix C for the visualization of this multi-stage generation procedure and the details hierarchy it forms.
>
> ---
>
> ### **W2, Q2. Limited Experimental Ablations**
>
> The added ablation studies have been updated in the new version of the paper, including:
>
> * unified model parameterization
> * use of interpolants
> * choice of resolution hierarchy (i.e., the stage count $K$)
> * the method by which the resolution embedding is incorporated
>
> ---
>
> ### **W2. Direct comparison between single-stage and multi-stage settings**
>
> Our comparison includes both single-stage methods—such as **SiD** and **SiD2** (direct pixel modeling), and **DiT** and **SiT** (latent-space modeling)—as well as multi-stage methods, including **PixelFlow** (direct pixel modeling) and **RIN** (latent-space modeling).
>
> ---
>
> ### **W3. Missing reference**
>
> We have updated the paper to include the studies you mentioned.
> Since these works introduce an alignment objective between the representations learned by the generative model and those produced by an external encoder, their supervision strategy is **orthogonal** to our proposed method.
> It would be an interesting future direction to combine such alignment techniques with CDC-FM.
>
> ---
>
> ### **Q3. Speed**
>
> The “Speed” metric in Table 1 is **seconds per generated image**.
> The inference configuration is provided in Appendix J.
>
> ---

---

> > ### Comment · Reviewer_Kt39 · 2025-11-25
> >
> > Thanks authors for their response and efforts to address all related concerns. I have adjusted the score accordingly.

---

> > > ### Author Response · Authors · 2025-11-25
> > > **Reply to Reviewer Kt39**
> > >
> > > We sincerely thank you for your constructive feedback and for taking the time to reassess our submission after the rebuttal. Your comments were valuable in helping us clarify our presentation and improve the overall quality of the paper. We appreciate your thoughtful engagement and will incorporate your suggestions in the updated version.
> > >
> > > Thank you again for your careful review and for contributing to the improvement of our work.

---

### Official Review · Reviewer_Wnzi · 2025-10-31

**Soundness:** 3
**Presentation:** 2
**Contribution:** 3
**Rating:** 6
**Confidence:** 2

**Summary:**

This paper presents a novel multi-stage generative framework based on stochastic interpolants, using a conditional dependent coupling strategy to address the computational efficiency-high-fidelity generation trade-off by structuring the process as a coarse-to-fine sequence of stages. Each stage uses an interpolant trajectory that transforms a structured prior, which is derived from the upsampled output of the preceding stage augmented with noise, into the target distribution. The framework ensures end-to-end optimization through a unified DiT model, enabling cross-resolution knowledge sharing while minimizing transport costs and inference time. Extensive experiments demonstrate that this approach achieves comparable or superior performance to existing generative models across multiple metrics.

**Strengths:**

1. The proposed solution, a unified multi-stage framework based on conditional dependent coupling, and coupled with a single DiT model, offers a solution that enables address the limitations of disjointed multi-stage approaches, while facilitating knowledge sharing across resolution stages and end-to-end optimization.

2. Extensive experiments to compare various methods are provided, demonstrating highly competitive generation performance.

**Weaknesses:**

1. The relationship and distinction between the proposed "conditional dependent coupling" and "data-dependent coupling" in the stochastic interpolant literature needs further clarification. A clearer discussion on how the work extends prior theory would help in precisely situating its theoretical contribution. Furthermore, the current theoretical presentation is highly technical, which may cause too much understanding burdens for a broader audience.

2. The depth of the experimental discussion is limited, maybe due to limited space, which has been taken by theoretical presentation. Since the experiments involve comparisons with numerous baselines across several metrics, the results and their implications are not discussed with sufficient granularity.

**Questions:**

My questions have been stated in the detailed comments.

---

> ### Author Response · Authors · 2025-11-23
> **Reply to Reviewer Wnzi**
>
> # reply to Reviewer Wnzi
>
> Thank you for the constructive feedback and thoughtful questions.
> Our responses to these points are provided below.
> **All corresponding revisions have been fully incorporated into the updated paper, colored in blue.**
>
> ---
>
> ### **W1. Relationship and distinction between the proposed conditional dependent coupling and data-dependent coupling**
>
> The proposed method introduces a unified multi-stage generative framework based on *conditional dependent coupling* to enable coarse-to-fine image generation.
>
> Data-dependent coupling, as previously established, shows that generating from a data-dependent (target-conditioned) source distribution to the target distribution is substantially easier than under standard independent coupling—yielding strictly lower transport cost, improved generative performance, and more efficient sampling.
>
> In contrast, the proposed *conditional dependent coupling* generalizes this idea to the multi-stage setting: it organizes multiple data-dependent couplings into a unified generative process, where each stage refines the output of the previous one in a coarse-to-fine manner.
>
> ---
>
> ### **W2. Limited experimental discussion**
>
> We have expanded the experimental section with additional studies, analyses, and discussions in the updated paper.
>
> ---

---

### Official Review · Reviewer_L5yh · 2025-11-01

**Soundness:** 2
**Presentation:** 2
**Contribution:** 2
**Rating:** 4
**Confidence:** 4

**Summary:**

This paper addresses the trade-off between computation and fidelity in existing image generation models (e.g., VAE-based latent space models suffer from information loss, pixel-space models are computationally expensive, and cascade models lack end-to-end optimization). It proposes **a unified multi-stage generative framework based on stochastic interpolants and a novel Conditional Dependent Coupling (CDC) strategy**. The framework decomposes high-resolution image generation into coarse-to-fine interpolant trajectories across K stages, all parameterized by a single Diffusion Transformer (DiT) to enable end-to-end training and knowledge sharing. Formal proofs (Theorem 2, Theorem 3) demonstrate that the method reduces transport cost and inference time compared to single-stage models. Extensive experiments on ImageNet-1k (256×256 and 512×512 resolutions) show state-of-the-art (SOTA) performance across metrics like FID (1.55–1.97 on $256\times 256$, 2.65 on $512\times512$), Inception Score (IS, up to 307.4), and inference efficiency (linear scaling with image size, Figure 2a). The source code is promised to be public upon acceptance.

**Strengths:**

1. **Novel Conditional Dependent Coupling for Accurate Distribution Learning**
    1. The CDC strategy ensures each stage k learns the accurate target distribution, unlike baseline multi-stage methods that target noisy, semantically vague distributions.
    2. The joint distribution formulation establishes conditional independence across stages, enabling a Markov chain inference process that avoids the computational complexity of autoregressive long-history conditions.
    3. CDC reduces transport cost by leveraging structured priors (upsampled low-resolution images with noise) instead of unstructured Gaussian noise.
2. **Unified DiT Design Enabling Efficiency and Knowledge Sharing**
    1. A single DiT parameterizes all stages, eliminating disjoint modules and enabling end-to-end training.
    2. Resolution embeddings (sinusoidal encoding of feature map resolution) enable the DiT to adapt to different stages without retraining.
    3. The unified design simplifies CFG integration, as a single CFG strength $S_\text{cfg}$ works across all stages.
3. **Comprehensive Empirical Validation Across Metrics and Resolutions**
    1. CDC-FM shows competitive performance of 30+ baselines on ImageNet-1k/256, with FID=1.81 (comparable to SiD2’s 1.72) and faster inference (Speed=1.31 vs. MAR-H’s 1.23). This demonstrates competitive fidelity and efficiency.
    3. Robust performance at 512×512 resolution, showing scalability to high resolutions.
4. **Theoretical Rigor with Formal Proofs**

**Weaknesses:**

1. **Incomplete Theoretical Analysis of Key Mechanisms**
   1. The paper claims the unified DiT enables "knowledge sharing" across stages, but provides no formal analysis of how resolution embeddings $e_k$ facilitate this (e.g., gradient flow across stages, parameter reuse statistics). No direct evidence found in the manuscript. I can only find qualitative claims in Sec. 3.2. This limits understanding of why the unified design outperforms stage-specific models.
   2. Ambiguities in coupling definition and Markov structure. The joint distribution in Eq. (14) conditions all stages on $x^{(k)}$, yet inference uses a Markov chain over stage outputs (Eq. 16), which appears inconsistent unless justified by conditional independence (Eq. 15). However, the transition from full conditioning to Markovian generation lacks explicit derivation. In Eq. (17), $\rho^{(k)}_0(\hat{x}^{(k)} | \hat{x}^{(k-1)}_1)$ is defined using $U_k(\hat{x}^{(k-1)}_1))$, but $\hat{x}^{(k-1)}_1$ is ambiguous. Should it be $\hat{x}^{(k-1)}_1$ (the output of stage $k−1$)? This notation mismatch could confuse implementation. Also, the deterministic mapping $m_k (x^k_1) = U_k(D_k(x^k_1))$ (Eq. 9) assumes invertibility of downsampling, but neighborhood averaging ($D_k$) is not invertible; the impact of this approximation on coupling fidelity is not discussed.
   3. The diminish factor $\gamma$ (Eq. 20) is critical for balancing diversity and efficiency, but the paper does not explain why $\gamma$=2.0 is optimal (Figure 2b shows FID dips at $\gamma$=2.0 then rises) or how $\gamma$ interacts with other hyperparameters (e.g., noise scale $\sigma$). This impacts practical guidance for users.
2. **Limited Experimental Ablations**
    1. The paper lacks ablations of the unified DiT vs. stage-specific DiTs. Without this comparison, it is unclear whether the unified design truly improves knowledge sharing or if performance gains come from CDC alone.
    2. The choice of linear interpolant ($\alpha_t=1−\tau, \beta_t=\tau$) is assumed without comparison to alternative interpolants (e.g., cosine, linear with warmup) that may affect trajectory smoothness (Eq. 12; Sec. 3.2; p.4). This limits understanding of sensitivity to path design.
    3. The diminish factor $\gamma$ controls noise decay ($\sigma_k = \gamma^{-(k-1)\sigma}$; Eq. 20), but only $\sigma \in [1.0, 5.0]$ is tested (Fig. 2b); no analysis of its interaction with stage count K or resolution hierarchy is provided.
    4. Resolution embedding via sinusoidal encoding (Sec. 3.2) is described briefly; its necessity versus simpler alternatives (e.g., learned embeddings, no embedding) is not verified.
    5. Qualitative results (Figure 3, Appendix B) show class-conditional generation but lack side-by-side comparisons with baselines (e.g., DiT-XL/2, PaGoDA, FractalMAR) to visually validate "high fidelity" claims.
3. **Reproducibility and Practical Guidance Gaps**
    1. While Appendix G provides implementation details (Table 3), it does not report memory footprints or training wall-clock time per model/depth. This limits practical adoption and reproductivity of this paper.
    2. The CFG integration (Sec. 3.4) uses a single $S_\text{cfg}$ across stages, but the paper does not test how $S_\text{cfg}$ impacts different stages (e.g., whether early stages benefit from higher $S_\text{cfg}$ for diversity). This limits understanding of CFG’s role in the multi-stage framework.
    3. No analysis of failure cases (e.g., mode collapse, structural artifacts in high-frequency regions) is included.

**Questions:**

1. **What evidence confirms that resolution embeddings enable knowledge sharing across stages?** The paper claims the unified DiT shares knowledge via resolution embeddings
$e_k$, but no empirical (e.g., parameter activation heatmaps, gradient similarity across stages) or theoretical (e.g., bound on parameter reuse) evidence is provided. Could you add an ablation comparing the unified DiT to stage-specific DiTs (same total parameters) on ImageNet-1k/256, reporting FID, training time, and memory usage? Additionally, could you clarify how $e_k$ is fused with timestep embedding (e.g., cross-attention layers, concatenation) with a small diagram or equation?
2. **How does the Markov chain structure (Eq. 16) reduce error accumulation compared to non-Markov multi-stage models?** The paper states the Markov property (stage k depends only on k-1) avoids compounding errors, but no data supports this. Could you report FID per stage (e.g., FID at $k=1, k=2, ..., k=K$) for CDC-FM and a baseline non-Markov multi-stage model (e.g., PixelFlow)? Additionally, could you measure the dependence between stage $k$ and $k-2$ to confirm the Markov property holds in practice?
3. **What principles guide the choice of diminish factor $\gamma$ and stage-specific noise scales $\sigma_k$?** Figure 2b shows $\gamma$=2.0 optimizes FID, but the paper does not explain why. Could you add an analysis of how $\gamma$ impacts noise diversity across stages and how $sigma_k$ interacts with $\gamma$?
4. **Why CDC-FM only uses 10 epochs for IN-1K-256 while other models use 400-1400 epochs?** Table 3 (Appendix G) shows CDC-FM trains for only 10 epochs on IN-1K-256, while baselines in IN-1K image generation task train for 400-1400 epochs. Could you clarify why *significant* fewer epochs suffice (only 2.5\% training)? Is it due to faster convergence from CDC or other factors? Also, you state that the per-gpu batch size is 4. Hence, could you report the number of your total batch size and how much memory it consumes during training/inference?
5. In addition, the name of Table 3 must be a mistake because it is titled 'Complete Parameter Configuration for *PixelFlow*' and Figure 2 is missing because you do not put a main caption of your Figure 2 but only sub-captions (a)-(d).

Overall, the paper makes contributions to multi-stage image generation via CDC and a unified DiT. However, it falls short of a higher score mainly due to sufficient experiments/ablations and mathematical ambiguities in Markov Chain, which makes me feel less confidence about some claims in the paper. Addressing the questions above would strengthen the contribution and soundness.

---

> ### Author Response · Authors · 2025-11-23
> **Reply to Reviewer L5yh (weakness part)**
>
> # reply to Reviewer L5yh
>
> Thank you for the constructive feedback and thoughtful questions.
> Our responses to these points are provided below.
> **All corresponding revisions have been fully incorporated into the updated paper, colored in blue.**
>
> ---
>
> ### **W1.1. Knowledge sharing and parameter reuse**
>
> One of the key advantages of a unified DiT over stage-specific DiTs is parameter reuse. In a unified DiT, the resolution embedding $e_k$ is treated as a conditioning signal that tells the model, “you are now operating at resolution $d_k$.” Consequently, the same parameters are shared across all stages $k$, rather than requiring separate parameters for each stage.
> See Q1.
>
> ---
>
> ### **W1.2. Conditional independence**
>
> There is a fundamental misunderstanding: the joint distribution in Eq. (14) is *manually constructed* through
> $x_1^{(k)} = D_{K \to k}(x^{(K)}) \sim \rho_1^{(k)}(x_1^{(k)} \mid x^{(K)})$
> together with Eq. (10), which is implemented by Eq. (9).
> This construction explicitly guarantees Eq. (15), which expresses the Markov property, and yields Eq. (14).
>
> ---
>
> ### **W1.2. Ambiguity of $\hat{x}_1^{(k-1)}$**
>
> Indeed, $\hat{x}_1^{(k-1)}$ is the output produced by stage $k-1$.
>
> ---
>
> ### **W1.2. From full conditioning to Markovian generation**
>
> See Appendix E for detailed derivation. See Appendix C for the Markov Chain visualization.
>
> During inference, Eq. (16) reverses the direction of the data-construction process in Eq. (14), analogous to the forward/backward procedures in flow or diffusion models.
> In our setting, the conditional distribution
> $\rho_1^{(k)}(\hat{x}_1^{(k)} \mid \hat{x}_0^{(k)})$ in Eq. (16) (backward, inference)
> is the reverse counterpart of
> $\rho_0^{(k)}(x_0^{(k)} \mid x_1^{(k)})$ in Eq. (14) (forward, data-pair construction for training).
>
> Using the transitivity of $D_K$, namely
> $D_{k+1 \to k-1} = D_k \circ D_{k+1}$,
> and the definition
> $x_1^{(k)} = D_{K \to k}(x^{(K)}) \sim \rho_1^{(k)}(x_1^{(k)} \mid x^{(K)})$ (Sec. 3.1),
> the distribution in Eq. (17)—implemented by
> $\hat{x}_0^{(k)} = U_k(\hat{x}_1^{(k-1)}) + \sigma_k \zeta_k$—is identical to
> $x_0^{(k)} \sim \rho_0^{(k)}(x_0^{(k)} \mid x_1^{(k)})$,
> which is implemented via Eq. (9).
> This equivalence follows from the fact that $D_k(x_1^{(k)})$ is identical to $\hat{x}_1^{(k-1)}$.
>
> ---
>
> ### **W1.2. Invertibility of downsampling**
>
> We neither assume nor claim the invertibility of the downsampling operation anywhere in the paper.
> The deterministic mapping
> $m_k(x_1^{(k)}) = U_k(D_k(x_1^{(k)}))$,
> used in Eq. (9), is solely intended to remove high-frequency details that are designed to be generated at stage $k$.
>
> ---
>
> ### **W1.3. Hyperparameter**
>
> As we mentioned—and as you correctly observed—the diminish factor $\gamma$ in Eq. (20) governs the trade-off between diversity and efficiency.
> The optimal value $\gamma = 2.0$ is obtained empirically, as shown by the sweep over different $\gamma$ values in Figure 3.
> The interaction between $\gamma$ and $\sigma$ is provided in the updated paper.
>
> When the stage-$k$ noise level $\sigma_k$ becomes too small (which occurs when $\gamma$ is too large or the base noise level $\sigma$ is too small), each $x_0^{(k)}$ collapses toward a lower-dimensional subset embedded in a higher-dimensional space. Consequently, the induced distribution becomes concentrated on a lower-dimensional manifold.
> Conversely, when $\sigma_k$ becomes too large (resulting from a too-small $\gamma$ or too-large $\sigma$), $x_0^{(k)}$ contains insufficient information from the previous stage. In this regime, more ODE integration steps are required to produce high-quality samples.
>
> Choosing an appropriate $\sigma_k$ for each stage mitigates both issues.
> A small amount of Gaussian noise smooths the base density, ensuring it is well-defined over the entire ambient space, while still preserving sufficient information from the preceding stage.
> This allows each stage to refine its input effectively using fewer simulation steps.
>
> ---
>
> ### **W2. Limited Experimental Ablations**
>
> The added ablation studies have been updated in the new version of the paper, including the unified model parameterization, the use of interpolants, the choice of resolution hierarchy (i.e., stage count $K$), and the method by which the resolution embedding is incorporated.
>
> ---
>
> ### **W3. Reproducibility**
>
> Implementation details and the model configuration are provided in Appendix J.
>
> ---
>
> ### **W3. Constant $S_{\mathrm{cfg}}$**
>
> The impact of $S_{\mathrm{cfg}}$ toward different stages is evaluated and validates the effectiveness of using a constant $S_{\mathrm{cfg}}$ in the updated paper.
>
> ---
> **Please see the question part in another panel due to the word limit.**

---

> ### Author Response · Authors · 2025-11-23
> **Reply to Reviewer L5yh (question part)**
>
> # reply to Reviewer L5yh
>
>
> **Please see the weakness part in another panel due to the word limit.**
>
> ---
>
> ### **Q1. Ablation comparing the unified DiT to the stage-specific DiT**
>
> See W1.1.
> The experiment and corresponding results are included in the updated paper.
>
> ---
>
> ### **Q1. Resolution embedding**
>
> See Appendix F for details about resolution embedding.
>
> ---
>
> ### **Q2. FID per stage for CDC-FM**
>
> We report the FID of CDC-FM for each stage in Figure 6 of the paper.
>
> ---
>
> ### **Q2. FID per stage for other multi-stage methods**
>
> For non-Markov multi-stage models such as:
>
> 1. **PixelFlow** and **RIN** — where the target distribution of each intermediate stage (except the final one) includes noise whose level decreases as $k$ increases,
> 2. **VAR** — where each stage models only the incremental details,
>
> it is **not feasible** to compute per-stage FID scores.
> These methods do **not** produce fully denoised samples at intermediate stages, so evaluating FID at those stages is not meaningful.
>
> ---
>
> ### **Q2. Dependence between stage $k$ and $k-2$**
>
> In the paper, we state that each stage is **conditionally independent**, as formalized in Eq. (15) and derived from Eq. (14).
> See **W1.2**.
>
> Conditioned on $x^{(K)}$, the stages are mutually independent.
>
> To provide intuitive understanding, we visualize the output of each stage along with the details added at that stage in Appendix C.
> From this visualization, it is clear that our multi-stage model forms a **Markov chain**:
>
> > conditioned on the output of stage $k-1$, stage $k$ is independent of stage $k-2$.
>
> ---
>
> ### **Q3. Hyperparameter**
>
> See W1.3 for analysis of $\gamma$ and $\sigma$.
> The results and analysis are included in the updated paper.
>
> The noise level of each stage $k$, denoted $\sigma_k$, is computed from $\gamma$ and $\sigma$.
>
> ---
>
> ### **Q4. Reproducibility**
>
> The previous configuration table was exported directly from our debugging code bench.
> We have now updated it to include the **full experimental configuration**, as shown in Appendix J.
>
> ---
>
> ### **Q5. Typo**
>
> Thanks for pointing out the mistakes.
> We have corrected these typos in the updated version of the paper.
>
> ---

---

### Meta-Review · Area_Chair_3t4A · 2026-01-03

**Summary:**

This paper proposes a unified DiT framework with a Conditional Dependent Coupling mechanism, claiming advantages in knowledge sharing across stages and efficiency. The main concerns across all reviewers converge on issues of ‌theoretical rigor, empirical completeness, and clarity of presentation‌, which currently limit the paper's impact, reproducibility, and persuasiveness.

‌**1. Theoretical Underpinnings are Underdeveloped‌:** The core mechanisms—specifically, how resolution embeddings facilitate knowledge sharing and how the proposed coupling advances prior theory—are not formally analyzed or sufficiently differentiated from existing work (R1 & R2). The theoretical section is noted as being both incomplete and overly technical, hindering understanding.

‌**2. Empirical Evidence is Insufficient to Support Claims‌:** A major flaw is the lack of critical ablation studies. The central claim that the unified design (not just CDC) drives improvement remains unproven without direct comparison to stage-specific baselines (R1, R3). The contribution of individual stages and the multi-stage process itself lacks quantitative and visual analysis (R3). The extraordinary claim of 40-160x training cost reduction based on 10 epochs is presented without adequate justification or verification, raising serious doubts (R1, R3).

‌**3. Reproducibility and Benchmarking are Deficient‌:** The paper omits key practical details like memory usage and precise training times (R1, R3). Evaluation is incomplete due to missing comparisons with state-of-the-art methods (R3) and an unclear definition of reported speed metrics (R3).

**4. ‌Presentation and Clarity Issues‌:** The balance between theory and experiment is skewed, leading to a superficial discussion of results (R2). Technical inconsistencies, such as potential mislabeling in the appendix, further undermine confidence (R3).

The paper presents an interesting idea but is not yet acceptable in its current form. It requires a ‌major revision‌ to address the above fundamental issues.

**Reviewer Concerns:**

1. R1 had raised 11 weaknesses and 5 questions. Only part of these concerns were addressed.

2. The main concerns of R2 and R3 were addressed by the rebuttal.

**Reviewer Scores:**

Since only part of R1's concerns were addressed, this reviewer would not increase the rating score even if participated fully in the discussion. According to (rating 4, confidence 4) of R1, (6, 2) of R2, and (4/6, 3) of R3, I tend to recommend a reject for this paper.

---

### Decision · Program_Chairs · 2026-01-26

Reject